# An Efficient LMS Platform and Its Test Bed

**Sooyoung Jung [1,2] and Jun-Ho Huh [3,*]**

1 Department of Electrical and Computer Engineering, Seoul National University, Seoul 08826, Korea; sjung7@snu.ac.kr
2 Hanmi E&C Inc., Seoul 08826, Korea
3 Department of Software, Catholic University of Pusan, 57 Oryundae-ro, Geumjeong-gu, Busan 46252, Korea
* Correspondence: 72networks@pukyong.ac.kr or 72networks@cup.ac.kr

**Abstract:** In order to develop an e-learning system as a method of education that frees both the teacher and learner from the constraints of time and space, it is necessary to develop software and to build the network equipment required to operate the software. The most basic system consists of a web server, a database server, and a video server. However, these elements are vulnerable to both internal and external threats. As for the web, database, and video servers, it is possible to respond to such threats by operating more than two devices, but this inevitably increases the cost of building the equipment. Therefore, this study proposed the use of a cloud service, such as AWS (Amazon Web Service), in order to save on the costs of purchasing, installing, and operating the servers, as well as a service designed to strengthen security by employees or trainees who understand the internal situation of the training institute. In other words, this study proposed the development of an efficient Learning Management System (LMS) platform and proved its efficiency using a test bed over a period of three years. The major contribution of this study is that the design of the proposed LMS has been improved to provide a more efficient performance than the existing LMSs by surmounting the traffic overload problem often found in video services. This is achieved by utilizing a lesser number of servers and maintaining the balance of the loads. Also, the interface used for the system can be adaptable to most of the web servers as they support Java, Android, and HTML-based systems. As a cloud-based LMS, this system has been tested for its efficiency and effectiveness for a period of three years during which the results have been satisfactory.

**Keywords:** LMS; LMS platform; Efficient LMS System; test bed; e-Learning; UML; platform; cloud computing; fog computing; LMS Architecture; Computer Architecture; Software Engineering

## 1. Introduction

Currently, the education industry is undergoing a paradigm shift from teacher-focused education, which requires a specific time and place of learning, to learner-focused education based on a ubiquitous computing environment that is accessible anytime, anywhere. It is not a traditional collective teaching method, but rather an e-learning system, i.e., a future-oriented method of education that frees both the teacher and learner from the constraints of time and space [1–4]. The e-learning system can be divided into synchronous education, wherein teacher and learner meet in a set place at a set time, and asynchronous learning in which they set their schedule and engage in learning on a non-real-time basis or in mixed-type learning. While offline education is conducted within a limited time and space with a certain number of students, e-learning has the advantage of being able to provide customized learning of the same content to multiple learners. However, it does have certain limitations compared with collective learning, as it is more difficult to manage and supervise learners efficiently using this method. To cope with such a problem, this study proposes an e-Learning Management System (e-LMS).

A Learning Management System (LMS) is a system that supports and manages the teaching/learning of learners. To provide the type of learning desired by learners in cyberspace, it is necessary to prepare both the courses and the process for participation in the learning by teachers and students. In the course of actual learning following completion of the preparation process, the learning process of the learners is tracked, and their learning history is managed in order to provide personalized learning to each learner. As such, the main functions of the LMS are the classroom organization function, cooperative learning function, attendance management function, and bulletin board function, all of which are necessary for online learning.

To provide these functions online, various pieces of network equipment and LMS software are required, including, for example, network equipment such as an LMS web server that provides an interface between the learner and the LMS, an LMS database that stores information related to the users' learning, and an LMS VoD database that stores multimedia files such as voice/video files. The network equipment and its configuration can be changed flexibly according to the purpose of the LMS.

Therefore, this study aims to present an enhanced LMS configuration diagram that can improve the robustness and reliability of the LMS after designing and implementing a basic LMS configuration diagram. This study also presents an economically enhanced LSM configuration that suggests a more economical structure. This is followed by the proposal of an economically-enhanced LMS, as proposed by Hanmi E&C Co., Ltd, supplemented with additional security.

In other words, the main contribution of this paper lies in the design of a learning system proven to be more efficient than a conventional LMS and capable of solving the traffic overloading of video services using a small number of servers to apply load balancing. Moreover, this paper provides a solution for the interface to web servers with Java, Android, and the existing HTML, as well as the operation of the servers. In today's sophisticated network systems, various types of threats are being posed by those with malicious intent, not only from inside but also from outside. They usually target system components such as web server, database server, or video server. It is possible to prevent such threats to a certain degree by taking some precautionary measures but the cost of installing additional equipment or hiring security specialists can be quite high. Thus, a secure cloud service can be an alternative to such a problem for the IT/ICT companies or educational establishments because of its higher security level. Thus, this study focuses on a cloud-based Learning Management System (LMS). The proposed LMS underwent three years of testbed experiments and its effectiveness and feasibility have been confirmed.

The paper is organized as follows: Section 2 discusses the related research; Section 3 presents the development of an efficient LMS system and its test bed; Section 4 discusses the implementation of an efficient LMS for e-learning and mobile-based learning, and server operation; Section 5 discusses the performance evaluation; and Section 6 presents the conclusion.

## 2. Related Research

The emergence of e-learning has had a significant impact on the education industry. Many educational institutions, cyber universities, and universities already provide PC web-based e-learning. E-learning started with the aim of providing interactive content. However, most classes are provided in the form of one-sided teaching by a teacher; but as there is an increasing need for interactive classes due to the rapid spread of smartphones, mobile learning is also needed. Furthermore, the rapid development of social networks, cloud computing, and IT technology, along with changes in personal behavior have led to changes in the education environment [5–8]. Thus, there is an increasing need for smart learning that supplements the existing type of education with e-learning and remote education. It will provide the momentum for education, and smart learning content can make course signup, GPA control and report management by developing LMS in consideration of the basic content and appropriate additional learning resources for self-directed customized content supply, open, sharing and cooperation [9–12]. An LMS is often used as a means of organizing university

learning administrative tasks. Of course, a learning management system is a system for managing and developing various processes of learning. The ultimate result of a learning management system can be connected to an academic management system. The LMS also includes the concept of academic management. For example, it allows the making of courses, registration, and checking attendance in cyberspace. However, it is not designed for academic management [12–15].

Meanwhile, an LCMS (Learning Content Management System) is a learning object management system that provides a function for managing the learning content uploaded to the LMS. In this regard, management includes the uploading of learning objectives, as well as their modification or deletion. It also provides a function for finding the desired learning object for reuse. In a system where an LCMS is not implemented, file-based content management is performed. A CMS (Content Management System) is a system for managing html on web servers, including tools for creating the html. An CMS system is also applied in various areas with diverse functions, but generally, it is not related to an LMS. LMS functions include content creation and file system uploading of content. As such, it may include some CMS functions [16–20].

However, an LMS is a client service-based database system that uses a web environment. An internet web environment is advantageous as there is no need to install a specific program for the service, as it can access the web through an internet browser. Recent e-learning methods have also used the merits of the web to the fullest possible extent. However, existing LMS options depend on hardware. So far, web-based database systems have used a 3-tier client-application-server model. The 3-tier model includes a web server, DBMS, HTML web-based server, application program, and web browser client.

## 2.1. E-Learning Success Research

The more significant part of the existing research concerning e-learning focuses on the implementation of an LMS and evaluating its environments and the results often obtained from traditional face-to-face learning methods. The research conducted for this study discusses and analyzes the results from a series of e-learning contexts. Picocoli, Ahmad, and Ives [21] studied the learning process in an LMS environment and drew a comparison with a face-to-face method in primary IT skills learning.

The results did not reveal any meaningful differences in these different environments as far as the performance of the participating students was concerned, even though ones who went through the e-learning process reported better self-efficacy and lower levels of satisfaction. On the other hand, Zhang, Zhao, Zhou, and Nunamaker [22] claimed that those who had experienced e-learning produced better academic results. Chou and Liu [23] drew a similar conclusion, saying that they exhibited better learning performance with a higher level of satisfaction. From these studies, it is possible to assume that the context of an e-learning environment is another influential factor. Meanwhile, some other LMS studies focused on the effects of LMS on student performance.

The central part of this study focuses on the technology acceptance model (TAM) proposed by Davis, Bagozzi, & Warshaw [24], the TAM2 by Venkatesh & Davis [25], and the unified theory of acceptance and use of technology (UTAUT) by Venkatesh, Morris, Davis, & Davis [26]. As the relevant research for these models, Van Raaij and Schepers [27] found differences between the participating students regarding acceptance level and use of LMS based on the conceptual model of TAM, TAM2, and UTAUT, respectively. At the same time, Pituch and Lee [28] maintained that even though factors, such as perceived usefulness, influence the use of LMS, the most significant influence on the students was the characteristics of the system itself. Selim [29] identified eight essential success factors of acceptance of e-learning by the students as follows: the attitude of the instructor, his/her teaching style, the student's motivation, the student's technical competency, the interaction between students, ease of access to the technology, the reliability of the infrastructure, and the support provided by the university or institution. The Expectation Confirmation Theory [30] was also used to describe the effects of the LMS.

In this regard, Hayashi, Chen, Ryan, and Wu [31] observed that the perceived usefulness and satisfaction had a direct influence on continued use in an LMS context, while the level of satisfaction was affected mainly by the perceived usefulness, confirming that both factors were positively associated with students' expectations about the LMS. By combining TAM and the expectation-confirmation theory, Roca, Chiu, and Martínez [32] found that students' intention to continue with the learning mainly depended on their level of satisfaction, which was influenced by perceived usefulness and confirmation. In addition to this observation, they also recognized that the service quality, system quality, perceived ease of use and cognitive absorption had a considerable influence on their level of satisfaction. However, although such factors may affect the use of LMS and e-learning, it is not clear how they are associated with the learning process itself. Only a few researchers considered this issue in the past, carrying out their research in the area of online collaborative learning.

According to Swan [33], the clarity of design, interaction with the instructor, and active discussion during the class influenced the students' perception. In this respect, Arbaugh and Benbunan-Fich [34] maintained that the collaborative environments seemed to be associated with higher levels of learner–learner and learner–system interactions, but only learner–instructor and learner–system interactions were significantly associated with an increase in perceived learning. Klobas and Haddow [35] observed that the students perceived that they were able to learn more and that their teachers also reported their increased learning when they participated in the collaborative learner–learner activities actively. Despite these findings, the studies conducted by these researchers do not account for all other e-learning systems. That is, even though the LMS offers functions that could be used to support collaborative learning, only a very few courses actually use them, so that many collaborative learning theorists do not agree that the results of online collaborative learning can be generalized to the environments in which e-learning is being used for material distribution or the support of unguided student interactions (Lipponen, Hakkarainen, & Paavola, [36]; Rudestam & Schoenholtz-Read, [37]).

It is true that a positive response to collaborative learning cannot be obtained evenly from all students. Through observation, Hornik, Johnson, and Wu [38] claimed that where there was a gap between a student's attitude toward the learning approach and the method implemented in an LMS, the student participated less in online discussions and was less satisfied with the learning process, leading to a reduced level of performance. For this reason, the most useful LMS research consists of studies conducted in a wide range of contexts with a variety of dependent variables along with a series of varying explanatory variables and models. As Coates et al. [39] aptly asserted: "It is difficult if not impossible to generalize from this research. The problem seems particularly acute when we try to understand the relationship between the context in which learning occurs, LMS use, and learning outcomes [40]."

### 2.2. LMS and Mobile Learning

Currently, the term "Mobile learning" is defined as a learning system that utilizes mobile devices including mobile phones, tablet PCs, and personal media players (Herrington & Herrington [41]; Valk, Rashid, & Elder, [42]) in either formal or informal educational settings (Quinn, [43]; Traxler, [44]). This form of learning has become popular due to the development of mobile devices with advanced wireless communication technology which support learning "on the move" in various educational settings, allowing students to access the learning content from different locations and times (Jones, Scanlon, & Clough, [45]; Hyman, Moser, & Segala, [46]; Garcia-Cabot, de-Marcos, & Garcia-Lopez, [47]) and to share the content with others (Woodill, [48]). Such an innovative method of learning encourages higher education institutions to aggressively adopt mobile technology to meet their students' expectations and needs. Currently, many undergraduate students bring their own digital devices to the class, especially small, portable ones such as smartphones and tablets (Dahlstrom, Walker, & Dziuban, [49]; O'Bannon & Thomas, [50]), expecting to access academic resources conveniently with them. Statistics show that approximately 5.5 million US students have enrolled in at least one online

course, accounting for around 25% of the total number of students who advanced to higher education institutions (Straumsheim, [51]). The figures also show that around 2.6 million students chose to attend online programs only.

Such a phenomenon is common to the international educational world (Hong-Wook Huh, Su-Ock Shon, Man-Kyu Huh, [52]). Universities and other higher education institutions are promoting the development of mobile learning management systems to support portable mobile devices, just as they did for PC-based LMSs. The basic functions embedded in their respective LMSs include conveniently accessing course materials and grades, sharing resources with other students or instructors, uploading assignments, and collaborating with classmates. Unlike PC users, current students are able to access LMSs at any time and from anywhere with rapidly developing mobile devices (Lowenthal, [53]).

Despite the current adaptation of formal LMSs in the mobile context, many researchers are still focused on the potential of informal mobile learning (Chen & Denoyelles, [54]; Hwang & Chang, [55]; Jun-Ho Huh, Kyungryong Seo, [56]; Martin & Ertzberger, [57]) and its application as an useful tool in a formal learning environment provided at least partially by other means (e.g., traditional online or face-to-face learning) (Gikas & Grant, [58]). Among a number of previous studies which addressed the issues pertaining to the provision of mobile learning as a tool to provide full access to academic, social, and administrative materials in the same way as a traditional LMS, very few researchers actually studied students' perceptions and behaviors when they were using this new technology (Kim, S.K.; Huh, J.H, [59]; Han & Han, [60]; M€odritscher, Neumann, & Brauer, [61]).

For instance, Cavus [62] studied learners' perspectives on mobile LMSs in higher education institutions, finding that the majority of students had a greater preference for the use of mobile devices when accessing their respective LMS. Furthermore, in studying the differences in perceptions between users and non-users of the mobile LMS, a notable study explained that while the students were aware of the advantages of the mobile LMS, some of them would not use the system because they felt that the system may be too complicated, making them psychologically resistant to participating in it [60].

On the other hand, others used the mobile LMS as they did not perceive the system to be too complex or challenging. Such differences may have arisen due to differences in their backgrounds or psychological development, or to some other external factors, but further research should be conducted on these factors. As such, it is important to determine which elements led these students to prefer a mobile LMS. Therefore, this study attempts to identify (1) the factors that affect the use of mobile LMS and (2) the correlations between those factors. Moreover, the study will identify the relationship between students' academic achievement and mobile LMS, although it is slightly early to accurately assess the results, as mobile LMSs have not yet been widely deployed at higher educational institutions (Cheon, Lee, Crooks, & Song, [63]; Hwang & Wu, [64]; Park, [65]). These issues should be addressed in future studies [66].

*2.3. Mobile LMS and Academic Achievement*

Currently, mobile devices are widely regarded as convenient and promising learning support tools which meet students' needs, encourage their participation in learning in unique ways (Hwang & Wu, [64]), and allow them to engage in educational activities (Wu et al., [67]). Despite such advantages, studies concerning the correlation between students' acceptance of mobile learning and their subsequent academic achievement are very scarce. The majority of studies have only focused on the factors that affect the user's inclination to use mobile learning (Cheon et al., [63]). The absence of empirical research or data related to this issue makes it difficult for researchers in the relevant subjects to identify the exact relationships (Hwang & Wu, [64]). There is another assertion that the use of mobile devices may distract or affect students' concentration (Gehlen-Baum & Weinberger, [68]). Nevertheless, mobile learning is still considered one of the most effective learning systems for its potential in terms of accessibility, flexibility, and ease of assessment and feedback, and the fact that it allows convenient access to online repositories and communities of practice (Jacob & Issac, [69]; O'Bannon & Thomas, [50]).

Again, further empirical studies are required to determine the effects of mobile devices on students' academic achievement Zydney & Warner, [70]. Mobile LMSs specifically provide students with unique opportunities to view lectures, participate in discussions, interact, and share ideas with others anywhere and anytime. Considering that the majority of students are cellphone or smartphone users, it would be possible for higher education institutions to provide these devices to their students in order to study the effects of mobile learning. Currently, it is possible to find at least one existing study on the structural relationships among individual, social, and systemic factors that have an influence on students' intention to use mobile LMSs and their influence on students' level of satisfaction with, and academic achievement in, online learning (Shin & Kang, [71]).

The many studies introduced at the beginning of this study rationalize the adaptation of the learning analytics dashboard. In addition to the above discussions, it will be helpful in studying students' interactions with one of the LMSs, namely Web CT, in two separate online courses. By studying and comparing motivation levels, it may be possible to identify the correlation between motivation and online participation (Hartnett, [72]). The results of the analysis of the data generated from the Web CT and the survey on the motivation of students involved in the first online course have yielded a statistically significant positive correlation between the motivation levels of learners and the number of messages posted by active participants (posts, hits, and reads). Also, the findings have proven that there is a relationship between the number of messages posted online and the assignment grades to active participants, as well as between the number of messages accessed by passive participants (hits and reads) and their assignment grades [73].

Navimipour & Zareie [74] claimed that the application of LMSs in online or remote learning courses was quite common in higher education, whereas Ashrafzadeh & Sayadian [75] observed that the integration of an LMS into learning and teaching practices had increased in the same areas. The studies pertaining to the motivation factors and the integration of instructional technologies reveal a possible correlation between instructional practices and the motivators (Gautreau, [76]). Besides student satisfaction, the level of satisfaction felt by instructors is also an essential factor affecting the adoption of the LMS by online courses. The issue here is how to measure and evaluate its effectiveness. For this, it is important to observe and understand instructors' experiences in developing their skills in utilizing online media and how it affects their satisfaction (Almeda & Rose, [77]). As LMSs are now widely used in the learning process, researchers should study the effectiveness of PC usage based on the measurements of user satisfaction levels with workplace PCs. This could be vital to the success or achievement of any programs or organizations (Bergersen, [78]; Del Barrio-García, & Romero-Frías, [79]).

Again, evaluating instructors' satisfaction is essential for improving classroom quality, as their use of an online platform to interact with their students without actually interacting with them in person leads instructors to be more cautious about using available online content (Swartz, Cole, & Shelley, [80]; Mclawhon & Cutright, [81]). Nevertheless, recognizing the limited formal uses of the LMS along with the wide range of LMS implementation could be critical for the future success of LMSs and their application (Naveh, Tubin, & Pliskin, [82]). However, many researchers such as Cigdem and Topcu [83] only focused on measuring instructors' acceptance of the LMS or their intention to use it in online courses, rather than on additionally measuring the instructors' level of satisfaction and the resultant outcomes of using such systems. By conducting both studies, it will be possible to gain a better understanding in terms of their preferences and intentions. As there are very few studies of this kind (Swartz et al., [80]) more attention should be paid to instructors' satisfaction (Vasilica Maria, Carmen, & Jose Luis Montes, [84,85]). This study focuses on developing an efficient LMS and a suitable test bed.

In A Learning Management System Enhanced with Internet of Things Applications, Mershad & Wakim [86] introduced an IoT-based learning management system (LMS) where various types of components, such as remote lectures, classroom monitoring, virtual reality, classroom experiment, security, classroom applications, student assessment, and data sharing, have been considered and

improved/upgraded. They expected that such an enhanced LMS would allow the learners, lecturers, and administrative personnel to have a positive experience while they participate in e-learning courses or performing administrative works. Their LMS model has considered the importance of efficiency and effectiveness of the system but also focused on the cooperations and interactions between all the parties involved in it.

In Trends and the Future of Learning Management Systems (LMSs) in Higher Education by Center for Educational Innovation (UB) [87], the three types of LMS's (i.e., proprietary LMS, open-source LMS, and cloud-based LMS) were compared and a similar comparison was made for the two major LMS vendors, namely Blackboard and Moodle. The literature also presented the future requirements for LMSs, emphasizing the necessity of enhanced LMS in higher education, efficient interoperation between LMSs, a deeper learning analytics, personalized experience in the LMSs, and a platform on which students and faculty are able to interact or collaborate smoothly.

Also, Cömert, Cömert, & Genc [88] focused on evaluating the technologies supporting the LMS's and introduced a unique algorithm called Adaptive Boosting which efficiently analyzes the learners' artificial behaviors and predict their future performances to make some suitable recommendations according to the analysis results.

Learning Management System (LMS) success: An investigation among the university students by Jafari et al. [89] studied the key elements/factors associated with the successful LMS models and established their own research model based on the analysis result obtained from reviewing the correlations between learners' performances and the effectiveness of the LMS's performances (e.g., provision of quality information/system, efficiency/effectiveness in system use and their level of satisfaction). The analysis results revealed that both the system and information quality were essential for a successful LMS, whereas the system use was strongly connected to the satisfaction level of the users.

Meanwhile, in an e-Learning management system using web services, Partheeban & SankarRam [90] developed and introduced an e-learning management system focusing on its web service-based framework which supports Cross Browsing and can be integrated with other types of databases. In this regard, their main objective was to perform a research on the methodology with which different types of platforms can be integrated, maintaining quality contents/leaning/delivery/evaluation management as well as secure access control.

Weaver, Spratt, & Nair [91] reported their findings from a survey which investigated the use of WebCT by the faculty and students through their learning and teaching experiences at one of the major Australian universities. The results obtained by distributing different types of questionnaires to the students and the faculty suggested that the students preferred the use of the technology (WebCT) and especially those who had been provided with well-established content and had established a good relationship with the faculty who were able to perform better with the technology showed a higher satisfaction level. On the other hand, the administrative personnel were more interested in the technical or administrative function of WebCT. Such findings have been useful for the Australian universities who constantly seek a better teaching environment for the students in terms of quality and efficiency.

Finally, in the Proactive e-Learning Management System, Zampuniéris [92] introduced a proactive LMS which helps the students to perform their online interactions more efficiently and effectively based on the functions provided in this LMS. The 'dynamic rules' applied to the system consist of the five rules for pre-condition, present condition, data collection, action, and rule generation, respectively. Table 1 shows a comparison of current systems.

**Table 1.** Comparison with Other Systems.

| Author | Title | Research Gap |
|---|---|---|
| Mershad & Wakim [86] | A Learning Management System Enhanced with Internet of Things Applications | As this literature review shows (as well as the table), there have been many studies regarding the technology used for learning management systems, and how it is evolving to be more "innovative". The trends for LMS technology are to make the system more interoperability, make it more secure, help the users and administrators be more collaborative, make the system more "smart" so less HR is needed to manage the system. However, there seems to be a gap on how cost-effective is to implement a LMS. In this paper, we hope to propose an innovative LMS that is far more cost-efficient than the current LMS. |
| Center for Educaitonal Innovation [87] | Trends and the Future of Learning Management Systems (LMSs) in Higher Education | |
| Cömert, Cömert & Genc [88] | A Study of Technologies Used in Learning Management Systems and Evaluation of New Trend Algorithms | |
| Jafari et al [89] | Learning Management System (LMS) success: An investigation among the university students | |
| Partheeban & SankarRam [90] | e-Learning management system using web services | |
| Weaver, Spratt & Nair [91] | Academic and student use of a learning management system:Implications for quality | |
| Zampuniéris [92] | Proactive e-Learning Management System | |

As with the above research works, many technologies have been studied to improve current LMSs, and the keywords for future LMSs are interoperability, security, collaboration, and smartness or intelligence, all of which make the system more autonomous and efficient. However, the issues pertaining to constructing such an LMS should be studied together while developing the technologies. Therefore, another new keyword for the Authors' future research work will be 'cost-efficiency'.

An upgraded and economically improved LMS design which guarantees system reliability is proposed in this research work. The framework of the design considers economic feasibility as well as the security, protecting it from all the possible threats. Aside from the economic aspect, this LMS model aims for the error-free performance during operation. After being tested by Hanmi E & C Co., Ltd for three years in the testbed, its efficiency was proved to be more efficient than any of existing LMSs in the Republic of Korea. One of the noteworthy features of the system is 'load balancing' which prevents system failure often caused by overloading.

The costs involved in constructing a VOD system for the lectures and linking the LMS system with learners' smart devices which allows them to participate in distance learning. These have been included in the total cost along with the software (customized) development cost, all of which are based on the expected price and labor wage index of 2018, as well as depreciation costs. The definition of the respective costs involved in developing individual functions are as follows:

- Functional Improvement Cost: Included in the maintenance cost, this is required when updating/modifying the homepage.
- Trouble Warning Service Cost: Also a part of maintenance cost. For troubleshooting the problems on the homepage or while providing the VOD service.
- Management Cost: A fixed cost where the costs of manpower, such as instructors' lecture fees and management personnel's wages, in addition to hardware costs (i.e., servicing, replacing, etc.).

The e-learning systems in today's environment require efficient and effective software and equipment to establish a network which allows the students and the lecturers to be freed from the limitations of time and space, offering more opportunities to them. Similar to other networks,

the e-learning network's basic structure requires servers which handle the operations on the web, and database and video functions.

However, vulnerabilities of the network and hardware still exist, and they are usually dealt with by backing up or by use of additional equipment, which can incur unnecessary costs. The utilization of a cloud service has become a feasible option with which the cost can be controlled. Although the utilization of a cloud service requires regular payments for the contract period, it could be a worthwhile strategy for e-learning providers as they can avoid or ignore other expenses associated with system operation.

The proposed service and the platform have been proved as efficient and effective through performance analysis over a 3-year operation period.

Meanwhile, approx. 60% cost reduction was achieved during the 3-year testbed experiment, compared to existing LMSs and the same experiment also confirmed the system's high integrity, efficiency, and error-free orientation. Also, its load balancing function was quite effective in preventing system failures resulting from system overload.

## 3. Development of an Efficient LMS System and Its Test Bed

Figure 1 shows the basic structure of the Learning Management System (LMS). Users access the LMS web server using a web browser. The LMS web server provides the VoD (Video on Demand) service according to the user's request and then allows the user to review and edit his or her information (user info), or provides the mobile web service according to the user environment. Besides, various functions required for online learning, such as notices, Frequently Asked Questions (FAQ), and online textbook purchases, are stored in a database (DB). Meanwhile, the LMS displays information that is useful to the learners, including external data such as weather information and online course news, as well as information provided by the LMS's own DB.

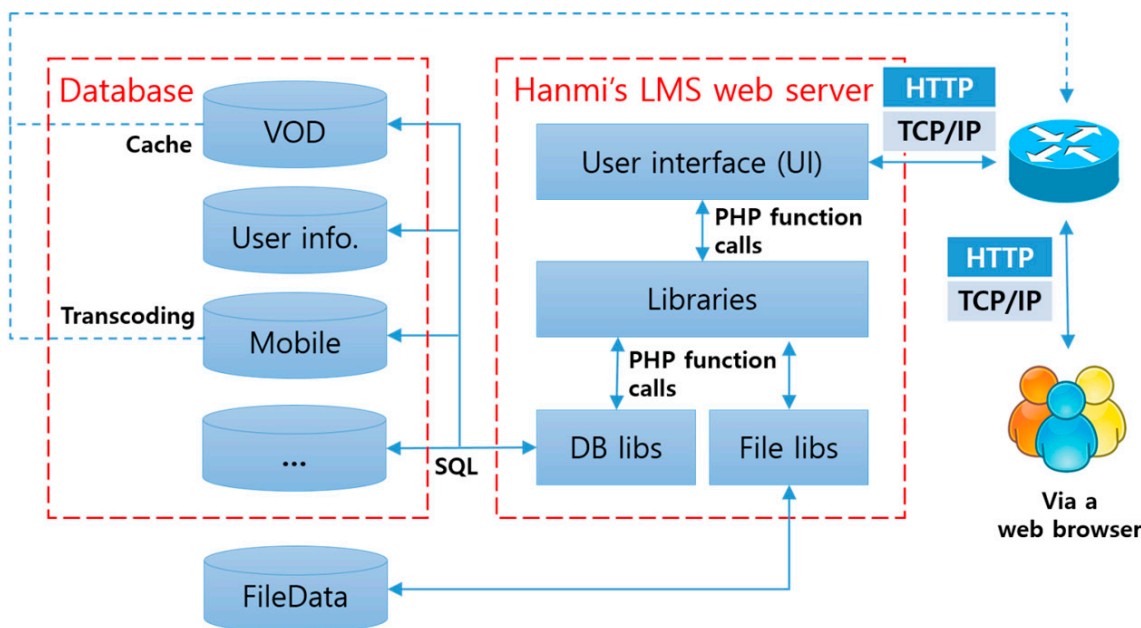

**Figure 1.** Basic learning management system structure.

Figure 2 shows the design of the network structure to provide the basic LMS services shown in Figure 1. First, the user accesses the LMS web server from a wired/wireless Internet access device using a web browser. The LMS web server analyzes the user's requirements and then delivers information that corresponds to those requirements from the LMS DB or LMS VoD server to the user. At this time, data about the requirements can be delivered to the user via the LMS web server or directly to the user without going through the LMS web server.

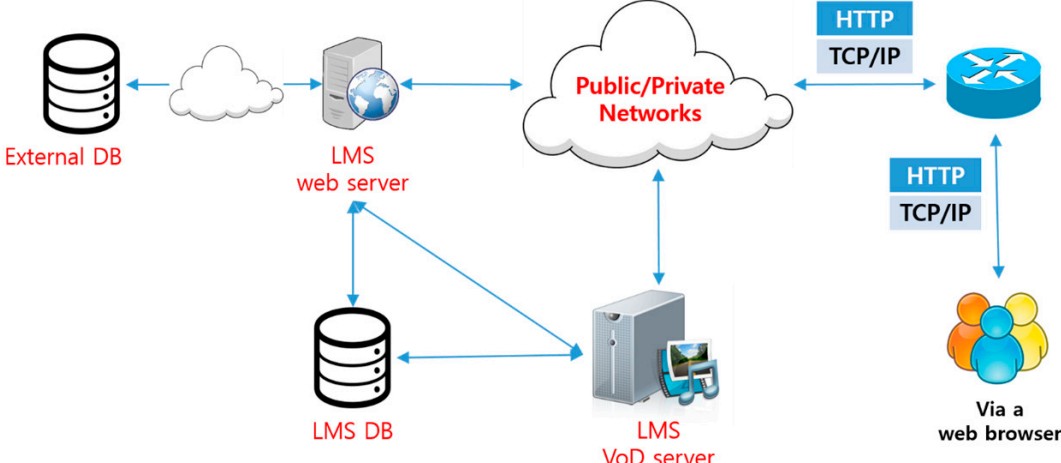

**Figure 2.** Design of the Network Structure.

As described in Figure 1, it is possible to provide a supplementary information service that is useful to users by interworking with an external DB. However, if the LMS system is configured as shown in Figure 2, the following problems may occur. First, there is the single point of failure problem. As shown in Figure 2, the LMS system consists of one LMS web server, one LMS DB, and one LMS VoD server. If anyone of these three network components stops working, the LMS service will become unavailable. If the probability of failure of each of the three elements is 30%, then the probability that the LMS system will not function properly is 90%. Second, since there is only one LMS web server, one LMS DB, and one LMS VoD server, there is no way to reduce the peak load time of users. To solve these two problems, this study proposed the following network diagram.

Figure 3 illustrates the proposed network architecture required to solve the problems with the network architecture designed for this study, as shown in Figure 2. The difference in Figure 2 is that the number of web servers, VoD servers, and DBs has increased to 2; and Web server 1, Web server 2, VoD server 1, and VoD server 2 are appropriately distributed by the load balancing function provided by the L4 (Layer 4) switch equipment. In other words, the L4 switch checks the load on the two web servers and the two VoD servers and sends the user's request message to the server with a lower load.

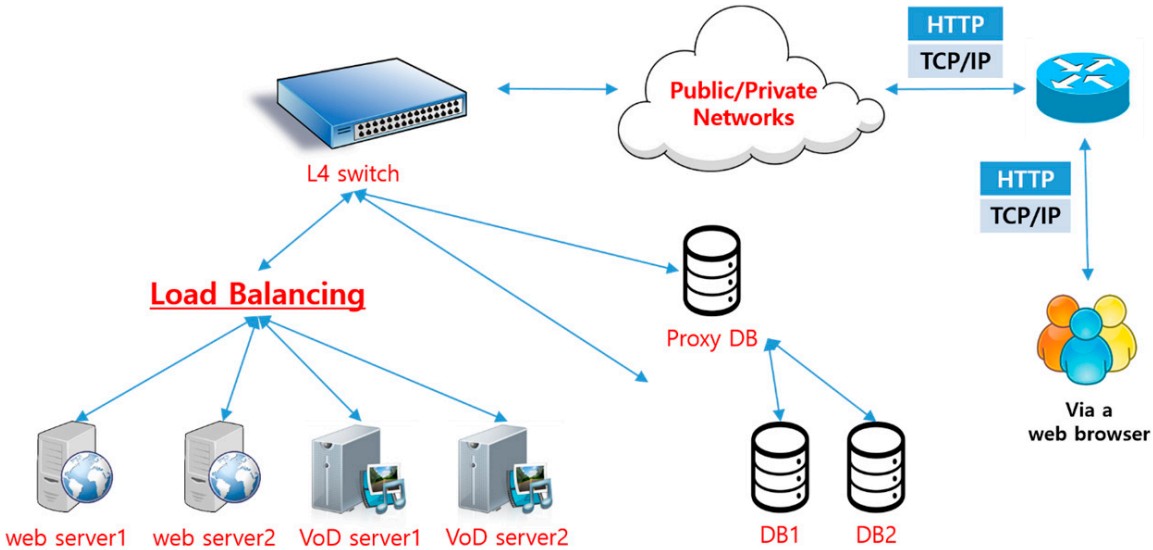

**Figure 3.** Proposed network architecture.

In Figure 3, two web servers and two VoD servers are designed, but each piece of server equipment can be increased to three or more depending on the size of the LMS system. Moreover, since two or

more web servers and VoD servers are installed, it can also solve the single point of failure problem. DB 1 and DB 2 are connected to the Internet via a proxy DB, which is a method of coping with equipment failure by duplicating DB 1 and DB 2. By connecting a proxy DB, rather than DB 1 and DB 2, to the Internet, the system can cope with an attack by a malicious user (i.e., a hacker). As shown in Figure 3, it is a robust system that is resistant to security threats, but it has a disadvantage due to the high initial purchase cost of at least ten high network devices. This high initial cost can be reduced by using a web cloud service, as shown below.

Figure 4 maintains the secure and robust system shown in Figure 3, and applies a cost-saving network structure. The changes represented in Figure 4 include the replacement of the L4 switch with a virtual L4 switch, and the option of replacing Web Server 1, Web Server 2, VoD Server 1, VoD Server 2, DB 1 and DB 2 with a web cloud service. Initially, the L4 switch server is purchased and replaced by the virtual L4 switch function provided by the web cloud service, which makes it possible to reduce the initial cost.

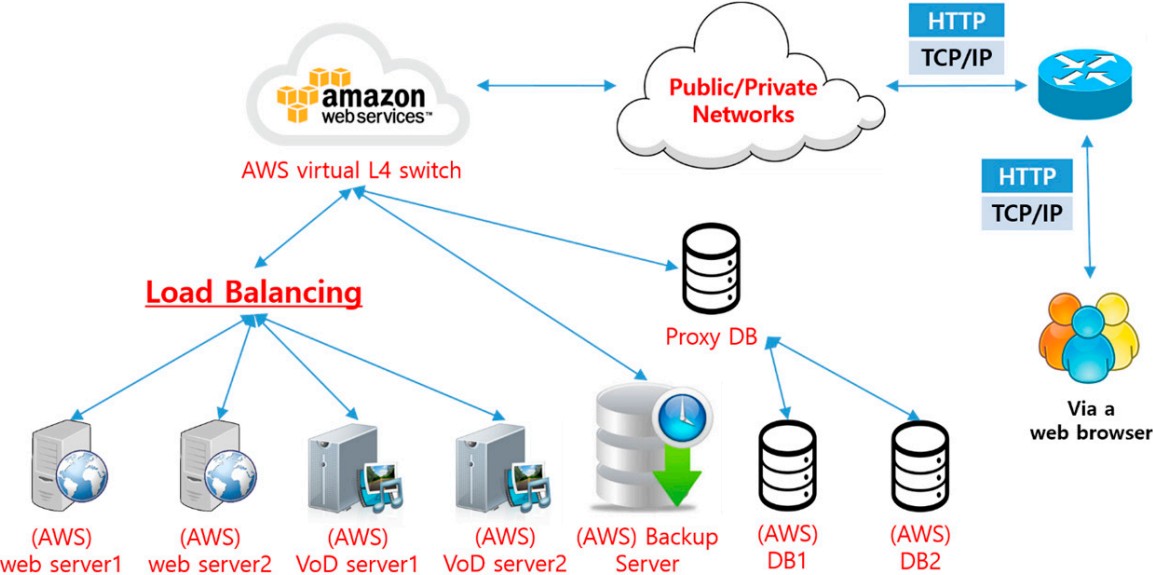

**Figure 4.** Maintenance of a secure and robust system.

Finally, Figure 5 shows the LMS finally provided by Hanmi E&C. It is a system that enhances the security of employees and trainees who understand the internal circumstances of the institute. In addition, securing a systematic operating system and expertise is most important in developing Korean-style Massive Open Online Courses (MOOCs). Also, it was necessary to define the roles and establish an operational strategy in order to induce the participation of universities and organizations. It was found that the integrated linkage of the existing Korea Open Course Ware (KOCW) and contents of ten centers, and the development of a system suitable for smart education would be needed to implement Korean-style MOOCs. For the purposes of this paper, a performance-enhanced architecture will be created by comparing the existing LMS and the K-MOOCS system. The information system of the K-MOOCS is an essential piece of information infrastructure. In addition, system performance and capacity control are very important issues because of the complexity of the system's configuration and the large-scale use target due to the characteristics of the K-MOOCS [4].

In other words, performance or management failures of the information system can lead to higher costs and waste of human resources, and can infringe on the student's right to take a class if satisfactory services cannot be provided. In addition, if a fatal problem occurs, such as a security incident, all K-MOOCS business could be paralyzed and other cyber colleges that share the content could be affected. Nevertheless, it is difficult to determine the adequacy of capacity because the hardware capacity of the information system should be estimated by considering all of the business characteristics,

the estimated work increase rate, the usage frequency of the users, and the characteristics of the development technology. This chapter describes the process of estimating the hardware size in detail and calculating the actual size by using the information system reference model of the virtual K-MOOCS, which has 1000 students, as shown in Figure 5.

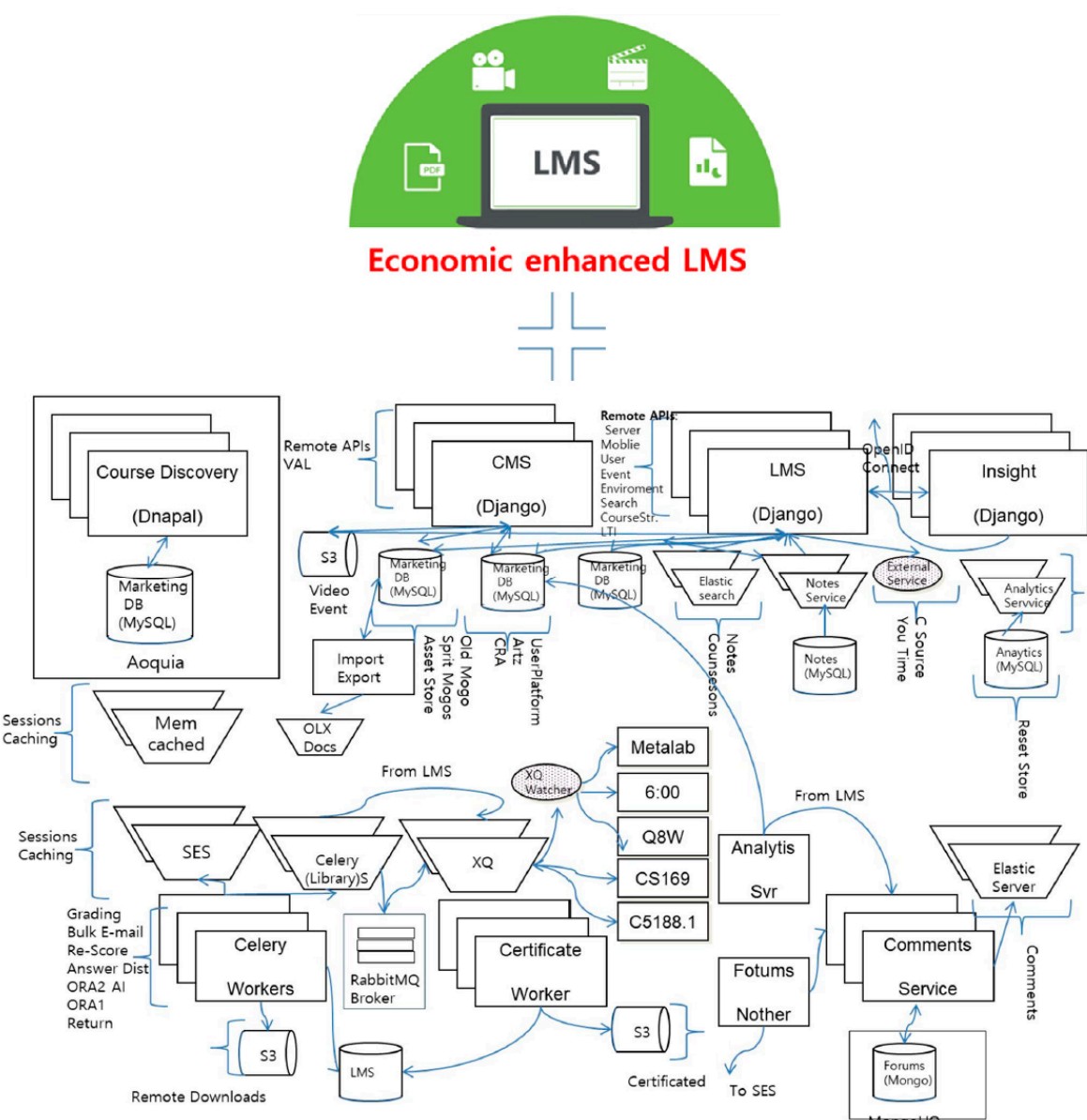

**Figure 5.** LMS to be finally provided by Hanmi E&C.

If the tuition payments, refunds, and partial refunds of offline/online students are processed manually, security problems will arise. In order to prevent this, an automatic management system service is provided by combining the payment, refunding, and partial refunding processes in the LMS system.

For example, if the cancellation and refunding processes are not connected, cases of internal fraud, such as refund without cancellation, may occur. To prevent such problems, the ultimate goal is to develop an LMS that minimizes staff intervention by allowing the LMS system to handle all the functions related to the attendance and payment functions.

## 4. Implementation and Test Bed of an Efficient LMS System for e-Learning and Mobile-Based Learning

Figure 6 shows the entire operating algorithm of the efficient LMS proposed in this study. The algorithm runs from the user's perspective and there are four functions in a single menu. The operation will return to the menu once each function has ended. The functions include Membership Registration, ID/Password Search, Register Course, and Take Class.

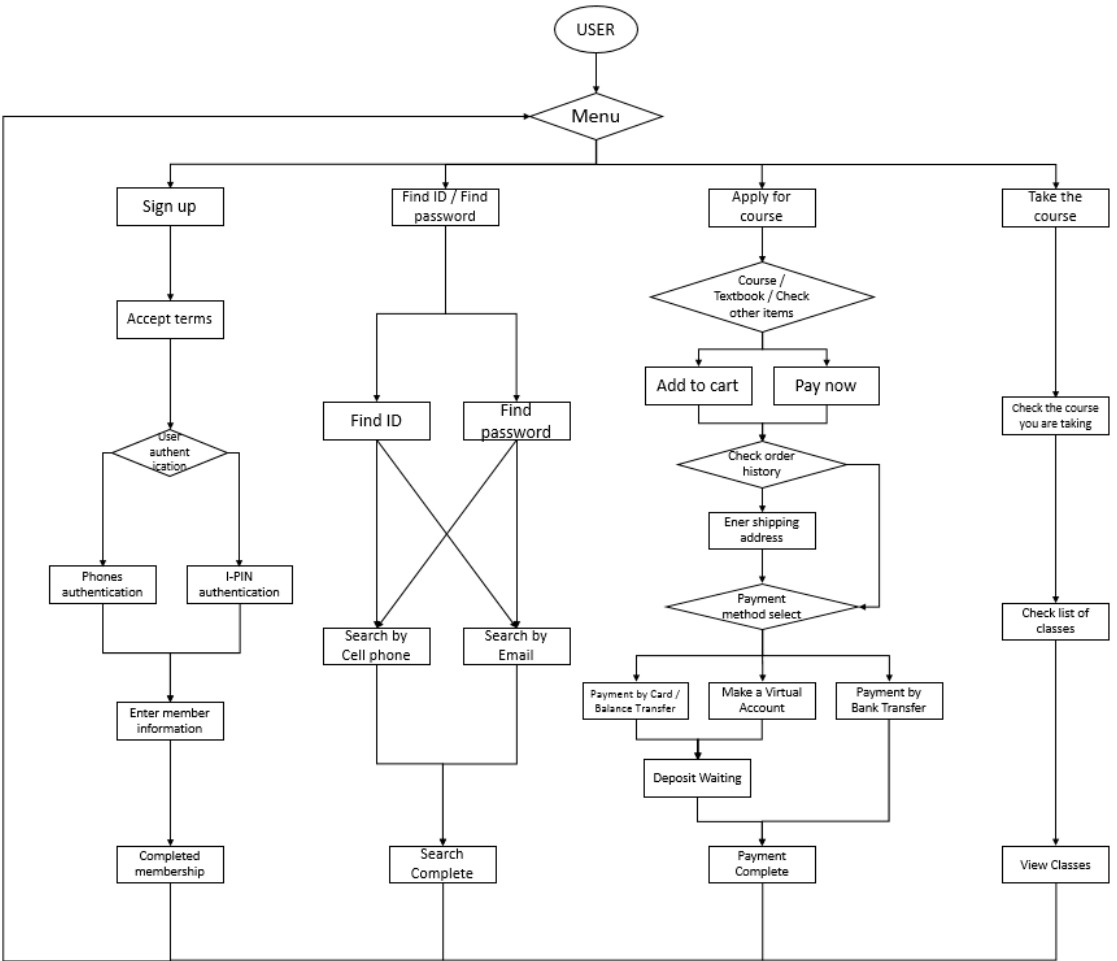

**Figure 6.** The entire operating algorithm of the proposed efficient LMS.

First, Membership Registration allows users to authenticate themselves in two ways after checking the courses they would like to select: via mobile phone or i-PIN. After authentication, they need to enter their information required for membership registration.

Second, the ID/Password Search function lets the users select either ID or password to be searched. The search result is then sent to either the user's mobile phone or e-mail depending on his/her choice, after which the operation will restart from Menu. In the third step, Register Course, the users need to check whether they can enroll in the course(s) they wish to attend. If confirmed, they should decide whether to send their selection to their basket or make payment followed by order checking. If they have already registered their shipping address, the screen will shift to another screen on which they can select the method of payment; otherwise, it will move to the screen where they are required to enter the shipping address followed by payment method selection. There are three ways to make the payment: by credit card/transfer, wire transfer, or through a virtual account. When the payment has been made by credit card/transfer or through a virtual account, the process will not proceed further until the payment is confirmed. Then, the process returns to Menu again. Finally, when Take Class

is selected, the users can check the course(s) they have selected and if there are no problems in the lecture list, they can attend the class online. The entire process is then completed and the users can return to Menu to select the functions they require.

Figure 7 shows a UML class diagram comprised of the following six classes: Study-taking class, ORDER payment class, Board-related class, Player image class, USR_Login class, and PROD related class. Figure 8 shows the UML class diagram.

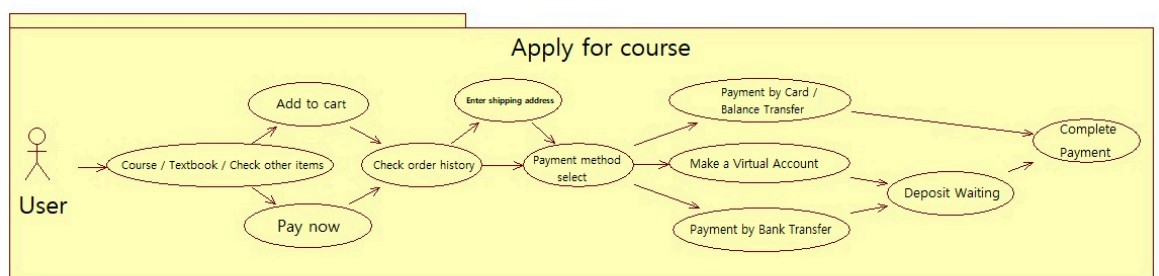

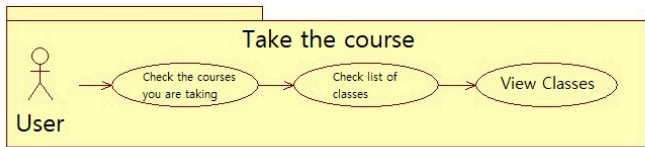

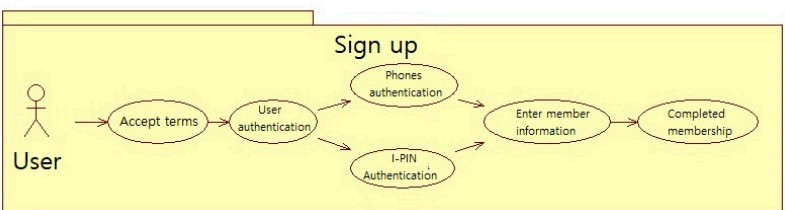

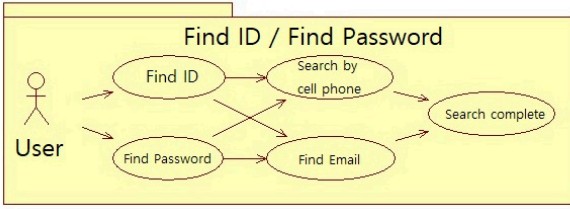

**Figure 7.** Use case diagram.

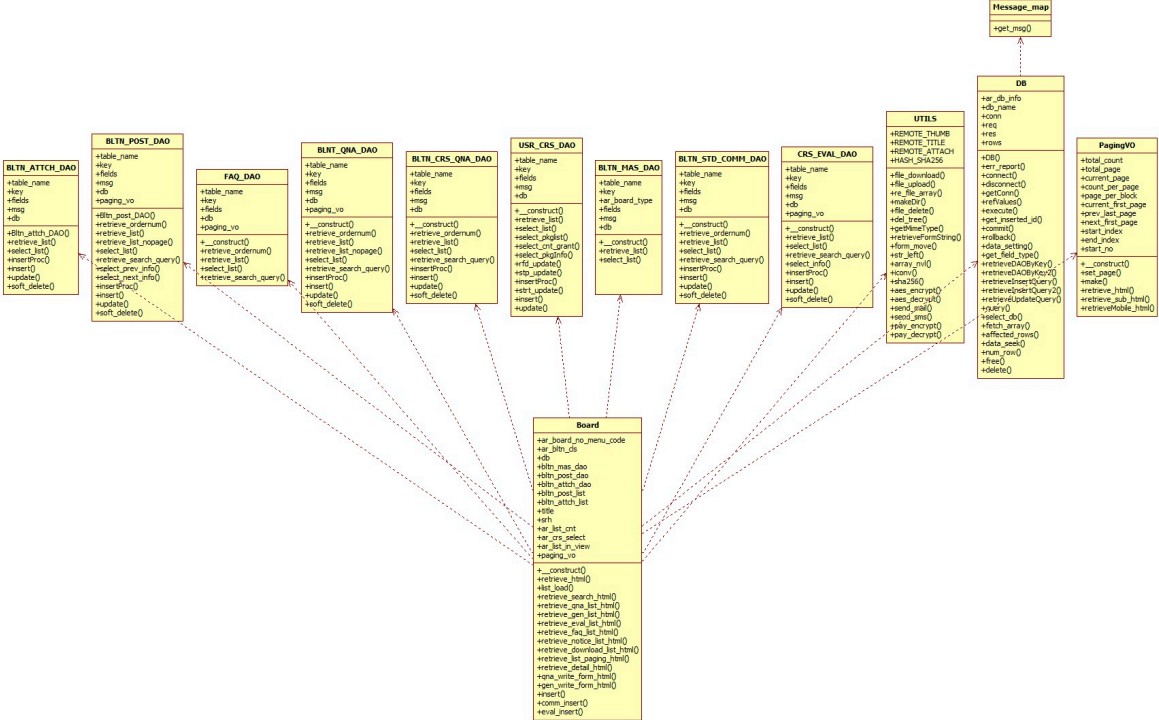

**Figure 8.** Unified Modeling Language (UML) class diagram.

The main parts of the study-taking class are enacted in an 8-part process. First, _construct allocates database objects to common variables. Second, retrieve() extracts study course list or detailed information data and returns it to html. Third, retrieve_detail() extracts detailed class taking information data and returns it to html. Fourth, retrieve_recent_lct() extracts recent take course information data and returns it to html. Fifth, retrieve_lct_list() extracts relevant course class list data and returns it to html. Sixth, retrieve_gen_crs_info() returns detailed general class information to html. Seventh, retrieve_pkg_crs_Info() returns detailed package class information to html. Finally, retrieve_list() extracts study course list data and returns to html.

Figure 9 shows the UML_ClassDiagram_ORDER. The main parts of the ORDER payment class are enacted in a 9-part process. First, _construct allocates DB object to common variables. Second, order_ck() checks the number of the product order. If it is an ordered product, it exposes message and process with page move. Third, retrieve_payList() extracts payment detail data and returns payment list to html character string. Fourth, retrieverfdList() extracts refund detail data and returns refund list to html character string. Fifth, retrieve_bskList() extracts shopping bag data and returns shopping bag saved list to html character string. Sixth, retrieve_list_paging() returns paging process data to html character string. Seventh, retrieve_sear ch() returns search layout to html character string. Eighth, retrieve_detail() extracts order detailed data and returns detailed order information to html character string. Finally, retrieve_rfdDetail() extracts detailed refund data and returns detailed information to html character string.

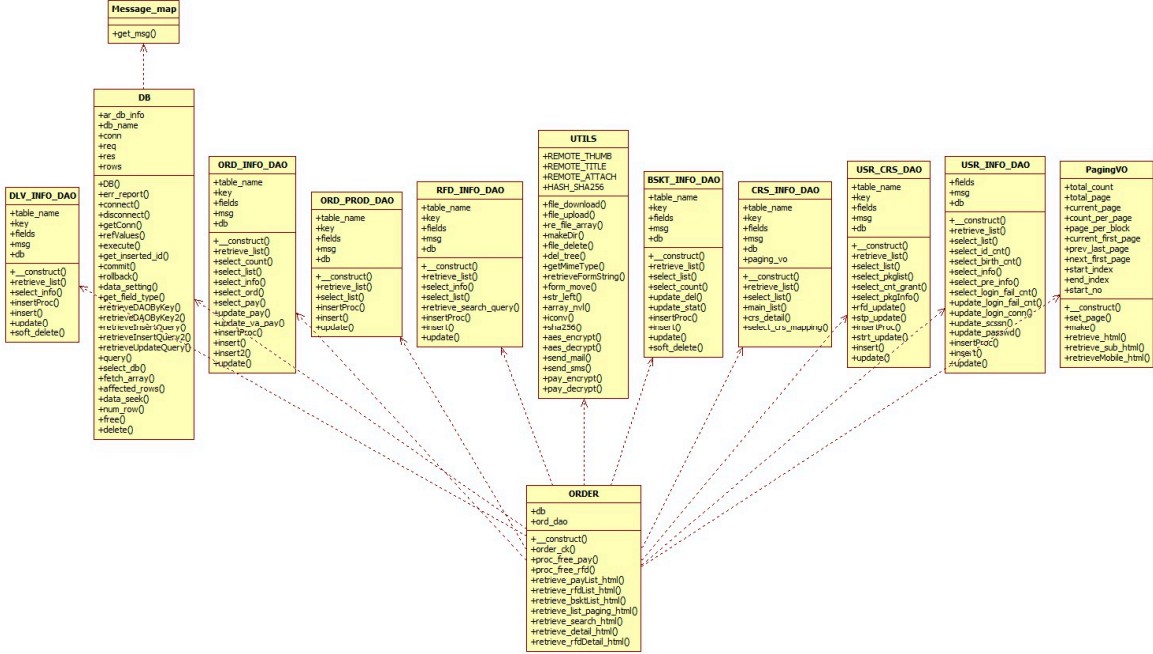

**Figure 9.** UML_Class Diagram_ORDER.

The board-related class is enacted in an 18-part process. First, _construct() allocates DB object to common variables. Second, retrieve() returns board list/detail/form to html. Third, list_load() extracts board data and saves it to common variable. Fourth, retrieve_search() returns search layout to html. Fifth, retrieve_qna_list() extracts Q&A board list data and returns it to html. Sixth, retrieve_gen_list() extracts common board list data and returns it to html. Seventh, retrieve_eval_list() extracts class review data and returns it to html. Eighth, retrieve_faq_list() extracts FAQ board list data and returns it to html. Ninth, Tenth, retrieve_notice_list() extracts notice board data and returns it to html. Eleventh, retrieve_download_list() extracts data room board data and returns it to html. Twelfth, retrieve_list_paging() returns paging processing data to html. Thirteenth, retrieve_detial() extracts detailed board data and returns it to html. Fourteenth, qna_write_form() returns Q&A board writing layout to html. Fifteenth, gen_write_form() returns common board writing layout to html. Sixteenth, insert() processes file upload and adds data to DB. Seventeenth, comm_insert() processes data to DB. Lastly, eval_insert() processes review data to DB.

Figure 10 shows the UML_ClassDiagram_PLAYER. The Player image class is divided into the following five parts. First, _construct() allocates the DB object to the common variable and extracts the relevant class information data. Second, contents() returns the class information and course information to html. Third, get_info() extracts the course information data from the DB and returns it to html. Fourth, view_player() saves the initial reproduction information and creates and returns the player screen to html. Fifth, save_bookmark() processes the bookmark information to the DB with revision.

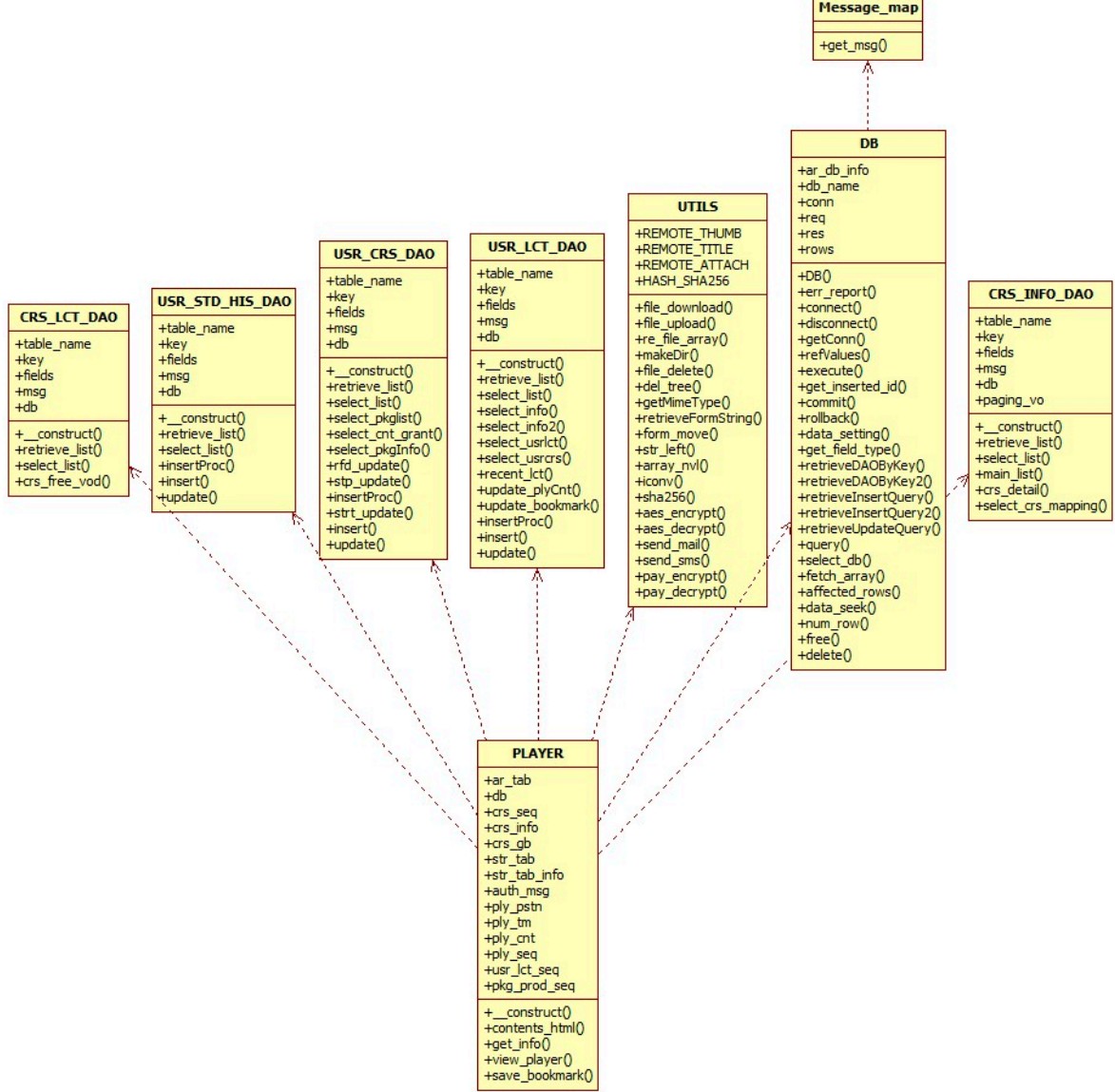

**Figure 10.** UML_ClassDiagram_PLAYER.

Figure 11 shows UML_ClassDiagram_USR. USR_Login class is divided into eight steps and class. First, USR_Login() initializes common variables. Second, checkPw() extracts member information data and processes error if the input password is incorrect. Third, checkUserInfo() checks password comparison/ID Korean check/ID character string. If there is no problem, and if there is no error message, true is returned. Fourth, login() extracts member information data. It checks data/log-in failure count. If there is a problem, process error is extracted. If there is no problem, it saves member information to the session for log-in and saves log-in history to DB. Fifth, secession() extracts member information data and compares password. If there is no problem, it saves secession information to DB and member information. Sixth, find() extracts member data that fits conditions, performs ID information return, issues a temporary password and sends the temporary password via email or SMS. Seventh, random_char() creates a character string randomly. Lastly, pw_modify() extracts member information data and compares password. If the password is correct, it modifies the password in DB. If not correct, it produces an error message.

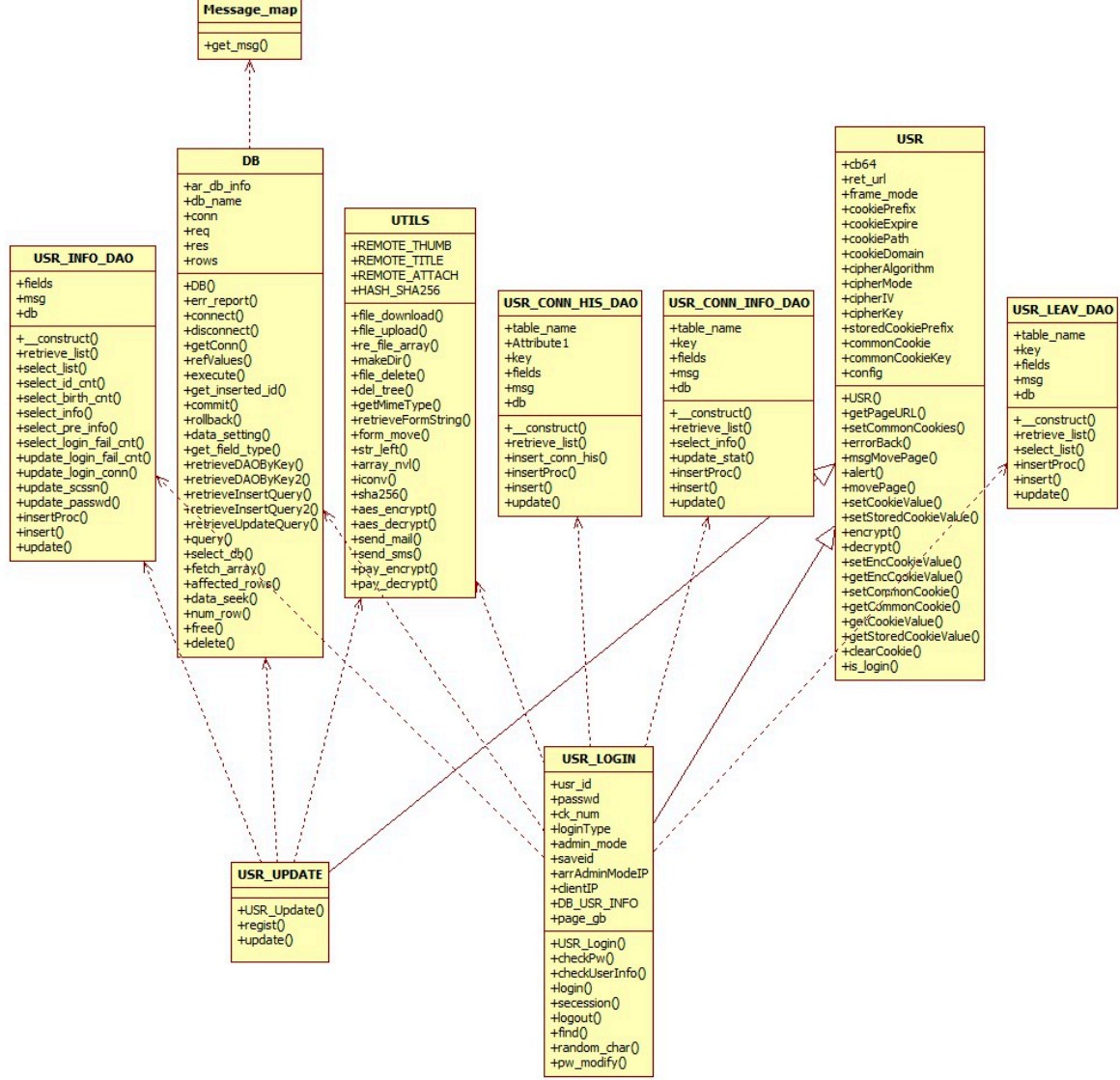

**Figure 11.** UML_ClassDiagram_USR.

Figure 12 shows UML_ClassDiagram_PROD. PROD class is divided into 14 classes. First, _construct() allocates DB object to common variables. Second, list_load() extracts data and save data into common variables. Third, retreive_grp_list() extracts product classification information data and returns it to html. Fourth, retrieve_pkg_list() extracts package class information list data and returns it to html. Fifth, retrieve_gen_crs_list() extracts class information list data and returns it to html. Sixth, retrieve_book_list() extracts textbook information list data and returns it to html. Seventh, retrieve_prod_list() extracts other product information list data and returns it to html. Eighth, retrieve_list_paging() returns paging process data to html. Ninth, retrieve_pkg_detail() extracts packaging process detailed information data and returns it to html. Tenth, retrieve_gen_crs_detail() extracts course detailed information data and returns it to html. Eleventh, retrieve_crs_lct_list() extracts relevant course information list data and returns it to html. Twelfth, retrieve_eval_list() extracts course review list data and returns it to html. Thirteenth, retrieve_bk_detail() extracts textbook detailed information data and returns it to html. Lastly, retrieve_etc_detail() extracts other product detailed information data and returns it to html.

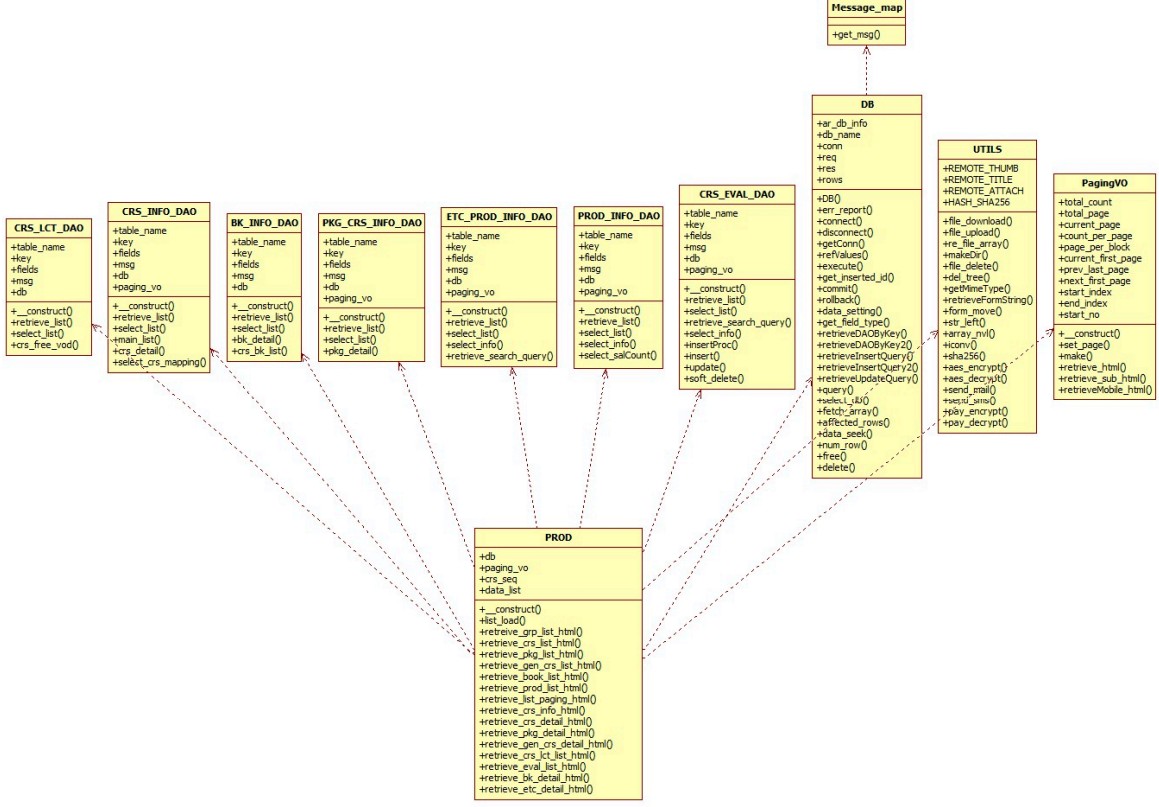

**Figure 12.** UML_ClassDiagram_PROD.

In the UML Sequence Diagram, the process of outputting class taking is as shown in Figure 13. The process consists of 28 steps, as follows.

When the algorithm is explained at the in-process class output, the User requires Member ID. Request it to USR. When receiving the value, input the value to getUSR ID() in user_id. User requests in-process class list or detailed in-process information html. There are two main functions of the algorithm. 1. Request detailed in-process class information, 2. Request in-process class list information. It is classified as an 'if' clause.

1. If the value of VIEW_MODE_DETAIL is $view_mode, it requests detailed in-process class information. As soon as the algorithm starts, general class data and package class data are requested from DAO. DAO executes SQL from DB to acquire data. General class data is saved in usr_crs_list variable, and package class data is saved in usr_pkg_list variable. Moreover, it requests detailed in-process class information html. If requested, it determines if there is any package in-process class data through 'if' clause.

(1) If there is package class data, request package class list and recently taken class data to DAO. DAO acquires and sends data from DB Package detailed class information, and the result value is requested for html using function retrieve_pkg_crs_info_html(). The existence of a package class is determined if there is any variable usr_pkg_list.

(2) If there is no package class data: Request general class list and recently taken class data as above. Then, acquire data and then request general detailed class information html. Each requested information requests html using function retrieve_gen_crs_info_html().

2. In-process class list information request is an option to request in-process class list information if the value of VIEW_MODE_DETAIL is not $view_mode. It is started by retrieving general class data and package class data from DB. General class data is saved into usr_crs_list, and package class data is saved into usr_crs_list variable. Request current in-process general class list using function

retrieve_list_html(). Then, the general class list is output as result value through repetitive statement as many as the number of the list. Then, the package class information is output in the same manner.

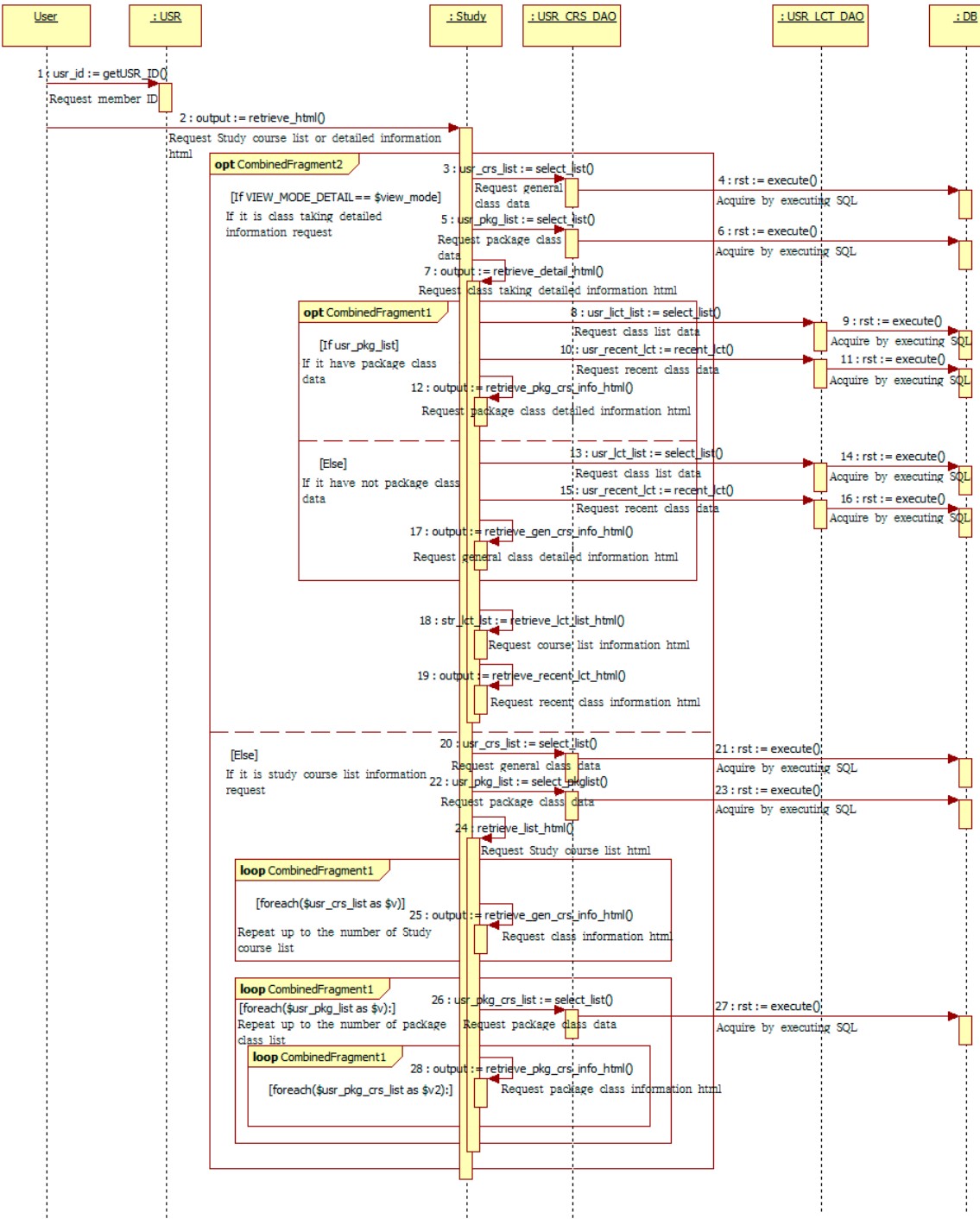

**Figure 13.** UML sequence diagram.

As a login procedure, the algorithm first asks the user to enter his/her member ID and then a process is initiated to request the information about the lecture schedule and their detailed starts after storing the html search value as a resulting value and moving to the Study page. opt CombinedFragment2 is divided into the case of requesting the detailed information about the lectures the user is taking with an If statement and the case of requesting the information about the list of

lectures the user is taking. Opt CombinedFragment2 is in the case of requesting the information about the lectures the user is taking.

A request is sent to the ':USR CRS DAO' page, which is an intermediate stage to import data from the DB after storing the user-selected general lecture list in 'crs list'. In the DAO page, the information is then requested and retrieved from SQL. Next, after storing the user-selected lecture package list in 'pkg list', a request will be made to the ':USR CRS DAO' page followed by the same procedure as above to retrieve the necessary data. Finally, an html search is conducted for the detailed information and the resulting value will be stored in 'output'.

Opt CombinedFragment1 is divided into the case where there is no lecture package or vice versa with an If statement.

For the former case, a request is made to ':USR LCT DAO' to retrieve a lecture list data. In this process, the data is retrieved with SQL and to obtain the recent lecture list, the data request is sent to DAO, after which the information is retrieved from SQL. Finally, an html search is conducted for the detailed information and the resulting value is stored in 'output'.

For the latter case, a data request is sent to DAO to store the user-selected lecture list in user_lct_list. In this process, DAO retrieves the data from SQL or if there is no lecture package, data of the user's recent lecture list is requested to DAO which will, in turn, retrieve it from SQL. A request for the detailed information on the general lecture courses is then sent to html to store it in 'output'. Next, the information on the lecture list is made to html for storing. Finally, a request for the recent lecture list is made to html for storing.

Opt CombinedFragment2: When a request is made for the lectures the user is taking, a request is made to USR CRS DAO page which is an intermediate stage for retrieving the data from DB after storing the user-selected general lecture list in 'crs list'. In the DAO page, information is requested and retrieved from SQL. Next, after storing the user-selected lecture package list in 'pkg list', a request is made to the USR CRS DAO page, followed by the same procedure as above to retrieve the necessary data. Then, the lecture list is requested to html. The 'loop CombinedFragment1' is repeated as many times as the number of the lectures in the list. Next, the information on the lectures is requested to the html repeatedly in the same way. The 'loop CombinedFragment1' is repeated as many times as the number of the lectures in the list as well. A request for the lecture package data is then sent to DAO where the data is retrieved from SQL for delivery. Finally, the information on the lecture package is requested to html while the loop statements are repeated as many times as the number of lectures in the lecture list.

Figure 14 shows the UML_ClassDiagram_STUDY. The UML Use Case Diagram is composed of four elements including class signup, class taking, member subscription, and ID and password finding.

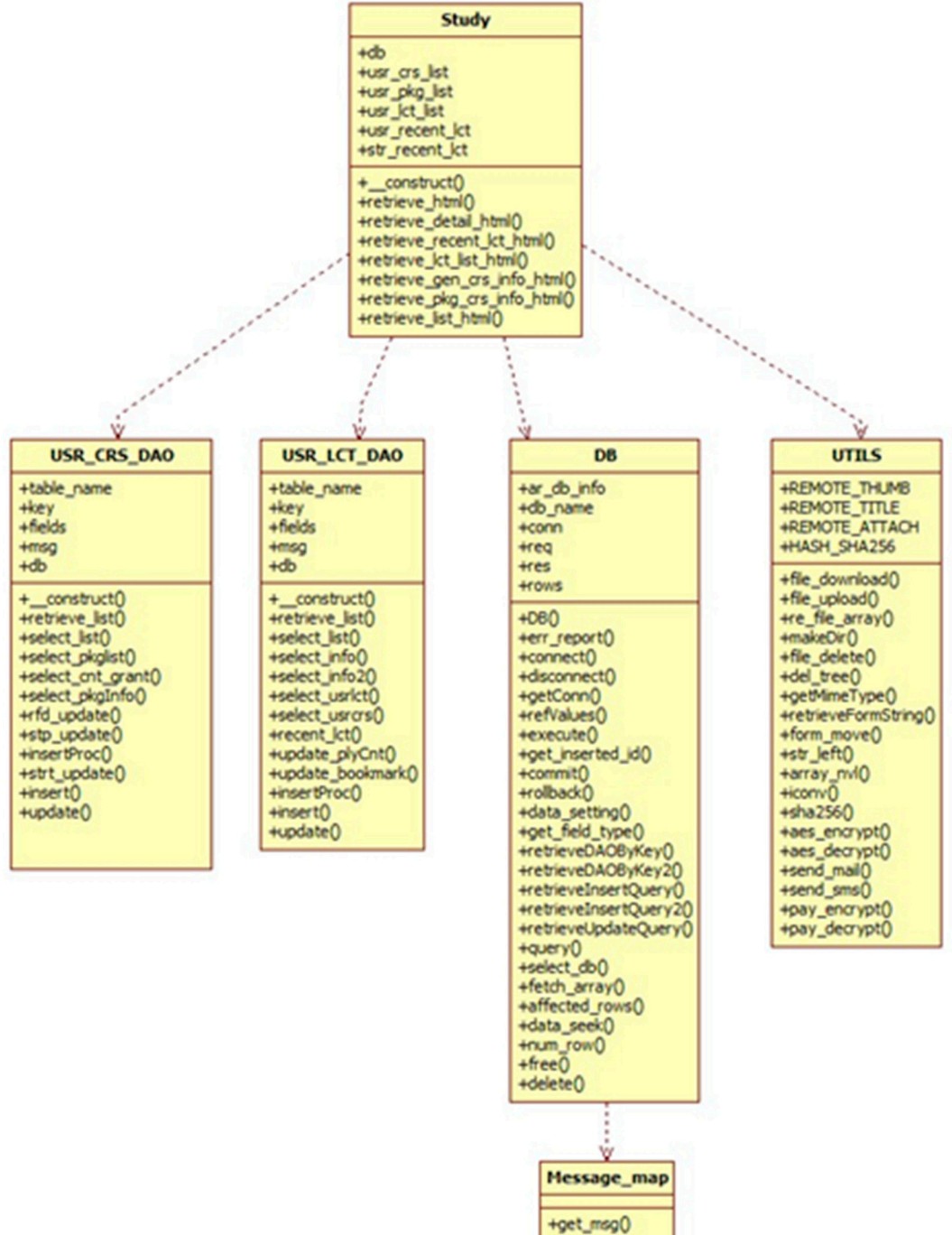

**Figure 14.** UML_ClassDiagram_STUDY.

In the class signup, if the user calls a page to check a class, the available class list page is displayed. Second, the user selects a class in the class list and clicks to put it in the shopping bag or to pay for the order at once. When "add to shopping bag" is selected, if the user clicks "order" on the shopping bag page, the order page is displayed. If the user selects "pay now", the order page is displayed. Third, when checking product information on the order page, and selecting the delivery place and payment method, the payment popup appears according to the type of payment means. When inputting payment information, the payment is processed. If a credit card is selected as the method of payment, the user inputs payment information and the payment is complete. If account transfer is selected as a payment method, it is transferred, and the payment is complete. If virtual account payment is selected, the virtual account is issued and standby status is made. Then, the complete

order page where the virtual account information can be checked appears, and the status changes to "Standby". Then, the order completion page appears in order to check virtual account information. After completing virtual account payment, the payment is complete. If non-bank note payment is selected as a payment method, the order status is changed to "standby". Then, the order completion page appears in order to check virtual account information. After completing virtual account payment, the payment is complete.

To take the course, the user logs in and the class taking page is called. Then, the page to check the study course list is called. Second, the user clicks one class title among the study course list and the view class page is called. Then, the page to check the course list is called. Finally, when the user clicks to listen to one lecture in the course list, the player appears as a popup, and the video is played for taking the course.

For member subscription, the member subscription page is called and then the "consent of terms and conditions" page is called. Second, the page checks consent for the terms and conditions and the privacy policy and calls the next step. Then, the identification page is called. Third, the user selects mobile phone certification or I-Pin certification in the identification. Then, the member information input page is called. Finally, the user inputs their member information and if it is complete, the complete page is called and the completion of member subscription guidance is simultaneously sent via email. ID/password finding is done using two methods. First, the user can call the ID finding page, where the user can select ID finding via mobile phone or email. When selecting ID finding via a mobile phone, the user needs to input the mobile phone number and information. Then, the page shows the searched ID. Second is to find the user's password. The user selects the password finding via mobile phone or password finding via email.

Figures 15 and 16 depict the actual configuration of the servers: Figure 15 shows two web servers, one DB server (master), one DB server (slave), and two DB backup video servers, while Figure 16 shows Web Server 1, Web Server 2, and Video Server 2, which can be load-balanced if necessary. Figures 15 and 16 show the server in operation.

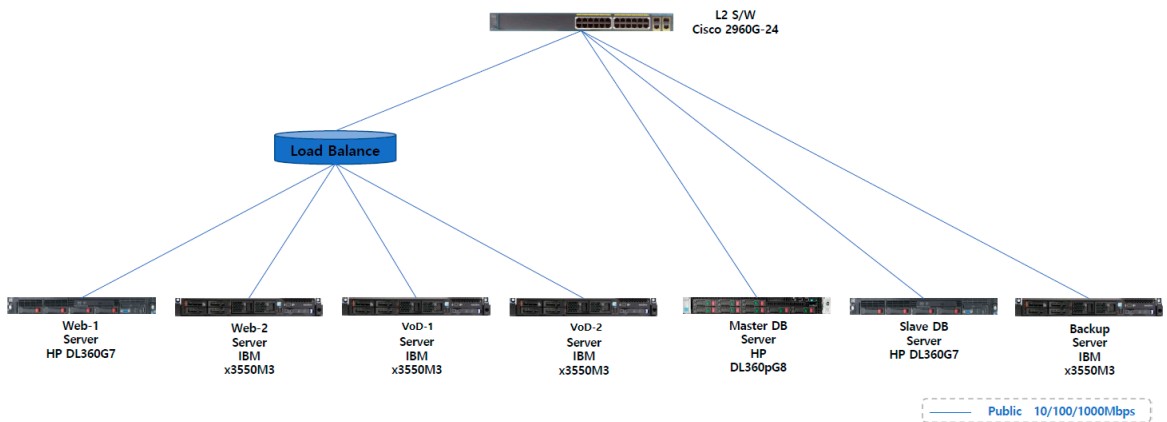

**Figure 15.** Servers in operation.

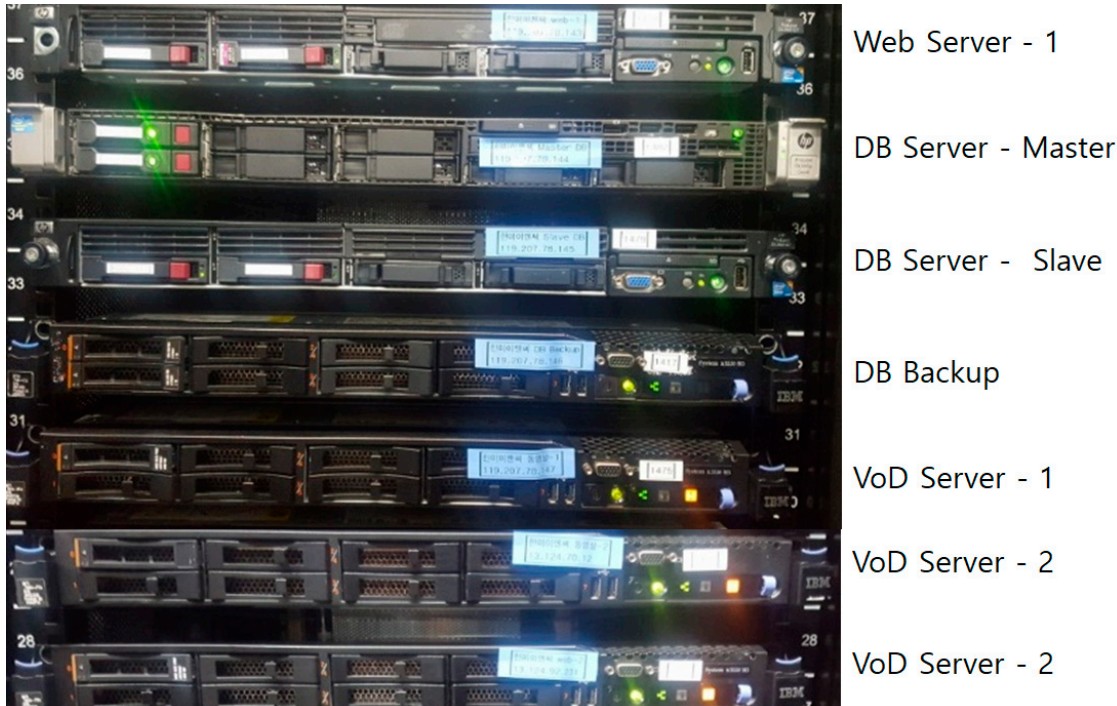

**Figure 16.** Network configuration.

Lastly, Figure 17 depicts the MySQL proxy configuration and process. First, the MySQL proxy is installed ahead of the DB server. Second, a client logs in to the proxy server. Third, the DB master server performs read and write. Fourth, the slave server performs read. Fifth, the proxy server performs distribution to the masters and slaves. Lastly, failover is detected in the proxy server. Maxscale from Maria DB was used as the proxy server.

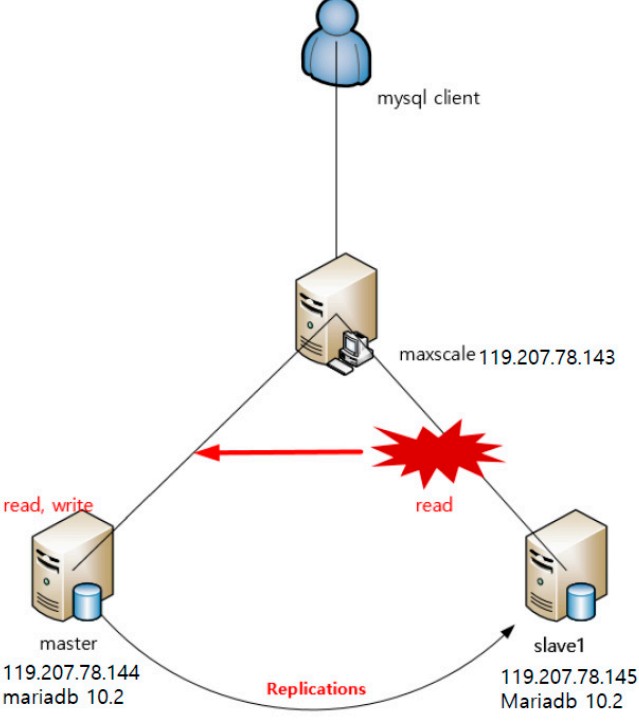

**Figure 17.** MySQL Proxy Configuration and Process.

## 5. Performance Evaluation

The system components, such as the web server, database server, and video server, are often vulnerable to threats from internal and external intrusions. Although the threats can be dealt with using the additional precautionary devices, adding the necessary equipment or devices to the system would require additional costs. Thus, for the employees/trainees of IT/ICT companies or the students of private educational institutes teaching network system management, use of a cloud service is suggested to avoid the costs of installing additional equipment or servers. A secure cloud service can guarantee the security of network systems while training these people. In this regard, development of an effective Learning Management System (LMS) is proposed in this study. The testbed experiments were conducted for a period of three years to prove the effectiveness and validate the viability of the LMS developed.

In the case of the Basic LMS, system maintenance costs of about four million won are incurred in the first, second and third years. It costs about 4 million won per year to maintain the LMS web server, LMS DB, and LMS VoD server, as shown in Figure 18.

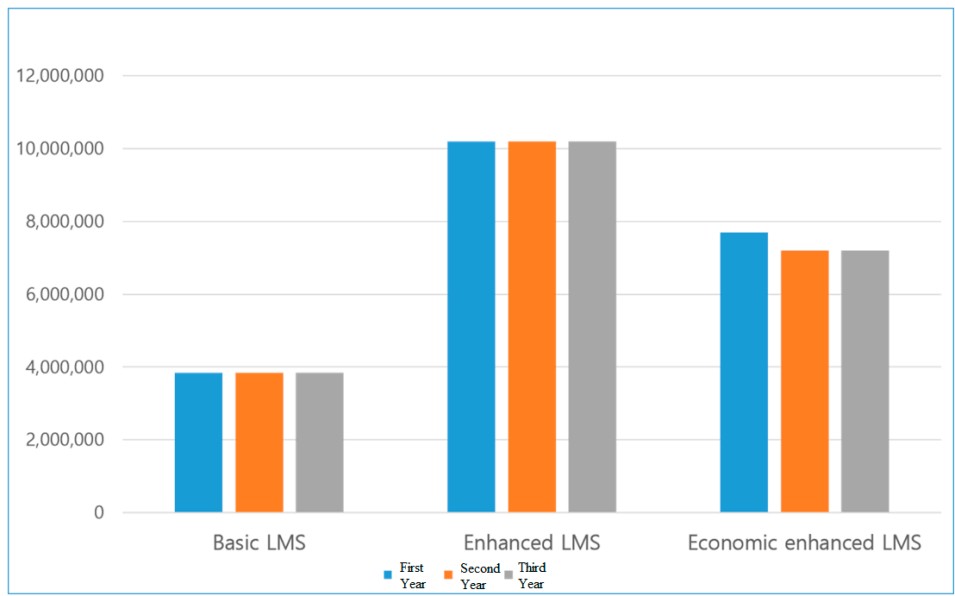

**Figure 18.** Graph comparing the cost of constructing the basic LMS.

Computer science and engineering studies often focus on methods of reducing time and costs. Table 2 compares the costs incurred between the existing LMS and the proposed LMS. Their respective costs calculated through testbed simulations were 10,200,000 Korean won and 3,900,000 Korean won, considering the involved costs such as homepage production, server management/operation, customized software, and other costs. The result clearly shows that the costs of the proposed LMS are much lower.

**Table 2.** The comparison of costs between existing LMS and the proposed LMS.

| Cost Unit: Korean Won | | | |
|---|---|---|---|
| **Production Costs** | | | |
| | | **Existing LMS** | **Proposed LMS** |
| Homepage | Domain | 50,000 | 50,000 |
| | Web Hosting | 200,000 | 200,000 |
| | Whole Design | 500,000 | 200,000 |
| Server Management | Web Server | 1,000,000 | 400,000 |
| | DB Server | 1,000,000 | 400,000 |

**Table 2.** *Cont.*

| Cost Unit: Korean Won | | | |
|---|---|---|---|
| **Production Costs** | | | |
| | | **Existing LMS** | **Proposed LMS** |
| Customized Software | VoD Server | 1,500,000 | 500,000 |
| | Backup Server | 500,000 | 250,000 |
| | Software Interlocking | 300,000 | 50,000 |
| | Mobile-Reactive Web Interlocking | 300,000 | 50,000 |
| | Interlocking with External Solutions | 300,000 | 50,000 |
| Other | VOD | 500,000 | 500,000 |
| **Maintenance Costs** | | | |
| Support & Maintenance | Functional Improvement | 350,000 | 100,000 |
| | Trouble Warning Service | 500,000 | 200,000 |
| Fixed Costs | Instructor (lecturer) | 1,100,000 | 400,000 |
| | Homepage Administrator | 800,000 | 200,000 |
| | Content Manager | 500,000 | 150,000 |
| | Server Cost | 800,000 | 200,000 |
| Total | Total | 10,200,000 | 3,900,000 |

It is undesirable to spend too much time on the subject of cloud computing before running into the discussion on the economics of the cloud system as we will definitely be dealing with the problem of 'CAPEX vs. OPEX'. For example, if a volume-rate service which adopts an external cloud system is used, the operating cost will be incurred continuously, but if the plan is to set up a data center autonomously, some investment has to be made for the facility. Thus, the comparisons have to be performed between facility investment cost and operating cost and there sure will be a controversy.

There have been many discussions on the cost comparison when 7X24 Amazon EC2 Instance is used for the server hosting in a certain company. Normally, it is customary that the average selling price of a 1U server is divided by 36 (typical expected lifetime of the equipment in months). After performing such calculation, the company concluded that their total operating cost per month was lower than the cost expected to be paid for the lease.

Based on such a result, people concluded that cloud computing can be more expensive than their own system and that it is inappropriate for the typical industrial applications which require 24/7 availability. However, the cloud system proposed in this study offers a service similar to Amazon but provides more efficiency based on the effective LMS platform developed for the employees. The enhanced LMS platform also focuses on the operational security and awareness of their internal situations. The cost efficiency has been proven by the system's error-free performance, offering a high availability during the 3-year operation.

Therefore, the cost of maintaining the LMS is much higher than that of the basic LMS. Specifically, it costs about 10 million won each year, i.e., the total cost of maintaining eight servers (about 7,200,000 Korean Won), plus the L4 switch fee (3,000,000 Korean Won). Compared to the Basic LMS, performance and stability are improved, but the cost is very high.

Finally, in the case of the economically-enhanced LMS, the L4 switch in the enhanced LMS is replaced by an AWS service. As an alternative to the AWS service, the cost of initial setup is about 500,000 Korean won, while the cost of the remaining eight servers is about 7,200,000 Korean won, as in the enhanced LMS. As a result, the cost in the first year is 7.7 million Korean won, but from the second year onwards, the server costs only 7.2 million Korean won. The economically-enhanced LMS shows superior stability and performance compared to the basic LMS and can be built at a lower cost than the enhanced LMS.

## 6. Conclusions

This study proposes an enhanced LMS structure that improves upon the robustness and reliability of the basic LMS system. Additionally, after designing the economically enhanced LMS structure, which is more economical in this particular model, the study proposes a secure, economically enhanced LMS structure that enhances security for the internal staff.

The cost for interlocking the system with smart devices for the users to attend the lectures via a browser or an application was added, as was the cost of developing customized software for that purpose, in addition to the cost of enabling the lecture VOD system. The cost calculation was performed based on the price and the labor price indexes of the first quarter of 2018, considering depreciation as well. Meanwhile, Functional Improvement included in the maintenance costs section refers to the cost of updating the homepage, whereas Trouble Warning Service is one that helps to solve the problems in the homepage or VOD service. The manpower costs for the instructors and management are included as fixed costs along with the server management cost, as this will be incurred continuously. From the testbed experiments conducted for a period of three years, it was possible to confirm that the proposed LMS had saved about 60% of the total costs required for the existing LMS on average.

Thus, the secure, economically enhanced LMS proposed herein is more secure than the basic LMS structure and is more robust in terms of reducing intermittent errors, as well as offering certain economic advantages. The system was serviced by Hanmi E&C Co. Ltd., and a test bed was performed for three years, thus proving that the proposed system is more efficient than the existing LMS system, and that load balancing solved the problem of system shutdown due to overloading.

In the meantime, MOOC and video lectures have recently become hot topics. As such, we expect that the proposed system will be adopted as a suitable system for schools with small servers and budgets, and as the base technology for e-learning and mobile-based learning. In the future, we plan to study large-scale services, such as flipped learning and K-MOOC, currently operated by universities.

**Author Contributions:** Conceptualization, S.J.; Data curation, S.J.; Formal analysis, S.J.; Funding acquisition, S.J.; Investigation, S.J.; Methodology, S.J. and J.-H.H.; Project administration, J.-H.H.; Resources J.-H.H.; Software, J.-H.H.; Supervision, J.-H.H.; Validation, J.-H.H.; Visualization, J.-H.H.; Writing–original draft, J.-H.H.; Writing–review & editing J.-H.H.

**Funding:** This work was supported by the National Research Foundation of Korea (NRF) grant funded by the Korea government (MSIT) (No. 2017R1C1B5077157).

**Acknowledgments:** This paper is a revised version of a paper entitled "Efficient LMS System Development and Its Test Bed for e-Learning and Mobile Based Learning" presented in the 2018 World Congress on Information Technology Applications and Services, 20–22 February 2018, JeJu, Republic of Korea [93]. Hanmi E&C provides video lectures of the US technology history to the world, has been providing video services for more than 10 years for national examinations for the public enterprises and secondary schools (middle school high school) in Korea and has designed an efficient model from the existing LMS systems. More details will be released in future contents.

**Conflicts of Interest:** The authors declare no conflict of interest.

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
