# Peer review of "An Efficient LMS Platform and Its Test Bed"

_electronics, doi:10.3390/electronics8020154_

Round 1
Reviewer 1 Report
After reviewing the article we have the following comments:
1. The main body of the article should clearly describe the contributions and the underlying principles based.
2. The experimental outcome should be compared with more literature, rather than just one, and outline its advantages and the disadvantages to make your research more valuable.
3. The detailed descriptions of the figures about the system had better be removed the to the appendix to make the article clear and understandable.
Besides, we found that figures 5 and 34 can be further improved both on the descriptions and on the figures themselves to persuade the reader, figure 20 and 21 are nearly the same, which might be problematic, and figure 4 and 33 are the same.

Author Response
After reviewing the article we have the following comments:
1. The main body of the article should clearly describe the contributions and the underlying principles based.
Thank you for your appropriate comment. The contribution of this research work has been newly added to the body, highlighting it in red. Thus, I respectfully would like to request your re-review if possible.
2. The experimental outcome should be compared with more literature, rather than just one, and outline its advantages and the disadvantages to make your research more valuable.
Your comment is highly appreciated. I’ve laid stress on the significance of the cost-related issues focusing on CAPEX vs. OPEX. The addition made to the study is highlighted in red as in the following.
Add) It is undesirable to spend too much time on the subject of cloud computing before running into the discussion on the economics of the cloud system as we will definitely be dealing with the problem of ‘CAPEX vs. OPEX’. For example, if a volume-rate service which adopts an external cloud system is used, the operating cost will be incurred continuously but if the plan is to set up a data center autonomously, some investment has to be made for the facility. Thus, the comparisons have to be performed between facility investment cost and operating cost and there sure will be a controversy.
Meanwhile, There were many discussions on the cost comparison when 7X24 Amazon EC2 Instance is used for the server hosting in a certain company. Normally, it is customary that the average selling price of a 1U server is divided by 36 (typical expected lifetime of the equipment in months). After performing such calculation, the company concluded that their total operating cost per month was lower than the cost expected to be paid for the lease.
Based on such a result, people concluded that cloud computing can be more expensive than their own system and that it is inappropriate for the typical industrial applications which require 24/7 availability. However, the cloud system proposed in this study offers a service similar to Amazon but provides more efficiency based on the effective LMS platform developed for the employees. The enhanced LMS platform also focuses on the operational security and awareness of the internal situations of them. The cost efficiency has been proven by the system’s error-free performance offering a high availability during the 3-year operation.
3. The detailed descriptions of the figures about the system had better be removed the to the appendix to make the article clear and understandable.
I appreciate your appropriate comment. The detailed explanations about the system-related numerical values have been moved to the newly arranged section (Appendix) to further enhance the readability of the study.
Besides, we found that figures 5 and 34 can be further improved both on the descriptions and on the figures themselves to persuade the reader, figure 20 and 21 are nearly the same, which might be problematic, and figure 4 and 33 are the same.
I really appreciate your attentive review and some of the points you’ve made for the problematic contents. An improvement was made for Fig. 5 & 34. Also, I thank you for correcting the mistakes made for similar figures (Fig. 20 & 21 and 4 & 33). The manuscript has been revised to avoid lengthy and somewhat meaningless explanations to achieve better completeness of the study. Therefore, I’d appreciate very much if you will review my study again if possible. The addition and changes made are being highlighted in red. Thank you.
The parts reducing the readability have been removed for improvement, especially Fig. 20 and 33.

Reviewer 2 Report
The paper provides a solid insight about the rationale for using LMS, presenting their advantages while providing a complete and comprehensive state-of-the-art on the subject (in some aspects it almost resembles a survey).
The english quality is good, but there are some aspects that do little to enrich the discussion and/or make ir more interesting. For instance, Figure 26 (which depicts the servers), as well as Figures 27-31 add little insight or value to the discussion - as such, authors should consider removing them.
The architecture of the proposed LMS system constitutes an interesting contribution, with special emphasis on the scalability and cloud component aspects. However, I cannot say the same for most of Chapter 4 contents, which are mainly focused on software engineering aspects that add little value to the reader, since they are specific of this particular solution. Also, the explanation for the mysql proxy configuration and process on page 29 is not easy to understand as it provides a sequential line of thought that hampers the clarity of the explanation,
Authors are advised to revise this paper, by focusing on the innovative aspects of the solution: the proposed architecture, the scalability and cloud component aspects. Focus on provisioning and management workflows is also important. Also, instead of providing flux and UML diagrams which mostly depict the functional aspects and supporting data models of the proposed solution, the authors should concentrate on improving the CAPEX/OPEX study providing a convincing argument for the proposed solution. Moreover, the testbed evaluation should focus more on replicating/proving the scalability and robustness benefits of the proposed solution.
Author Response
Comments and Suggestions for Authors
The paper provides a solid insight about the rationale for using LMS, presenting their advantages while providing a complete and comprehensive state-of-the-art on the subject (in some aspects it almost resembles a survey).
The english quality is good, but there are some aspects that do little to enrich the discussion and/or make ir more interesting. For instance, Figure 26 (which depicts the servers), as well as Figures 27-31 add little insight or value to the discussion - as such, authors should consider removing them.
The architecture of the proposed LMS system constitutes an interesting contribution, with special emphasis on the scalability and cloud component aspects. However, I cannot say the same for most of Chapter 4 contents, which are mainly focused on software engineering aspects that add little value to the reader, since they are specific of this particular solution. Also, the explanation for the mysql proxy configuration and process on page 29 is not easy to understand as it provides a sequential line of thought that hampers the clarity of the explanation,
First of all, I appreciate that you’ve read through my study entirely and giving me an appropriate comment. The parts reducing the readability have been removed for improvement, especially Fig. 27-31. Also, for the explanation concerning configuration and process of the MySQL proxy (page 29), some of the obstructing elements have been reduced and to further clarify the contribution of this study, the significance of the research was newly added.
Authors are advised to revise this paper, by focusing on the innovative aspects of the solution: the proposed architecture, the scalability and cloud component aspects. Focus on provisioning and management workflows is also important. Also, instead of providing flux and UML diagrams which mostly depict the functional aspects and supporting data models of the proposed solution, the authors should concentrate on improving the CAPEX/OPEX study providing a convincing argument for the proposed solution. Moreover, the testbed evaluation should focus more on replicating/proving the scalability and robustness benefits of the proposed solution.
à I’ve focused on the innovative aspects of the proposed solutions (architecture, scalability, and cloud-configuring elements) while laying stress on CAPEX vs, OPEX problem. With these improvements, I respectfully would like to request your re-review if possible. The additions and changes are being highlighted in red.
Add) It is undesirable to spend too much time on the subject of cloud computing before running into the discussion on the economics of the cloud system as we will definitely be dealing with the problem of ‘CAPEX vs. OPEX’. For example, if a volume-rate service which adopts an external cloud system is used, the operating cost will be incurred continuously but if the plan is to set up a data center autonomously, some investment has to be made for the facility. Thus, the comparisons have to be performed between facility investment cost and operating cost and there sure will be a controversy.
Meanwhile, There were many discussions on the cost comparison when 7X24 Amazon EC2 Instance is used for the server hosting in a certain company. Normally, it is customary that the average selling price of a 1U server is divided by 36 (typical expected lifetime of the equipment in months). After performing such calculation, the company concluded that their total operating cost per month was lower than the cost expected to be paid for the lease.
Based on such a result, people concluded that cloud computing can be more expensive than their own system and that it is inappropriate for the typical industrial applications which require 24/7 availability. However, the cloud system proposed in this study offers a service similar to Amazon but provides more efficiency based on the effective LMS platform developed for the employees. The enhanced LMS platform also focuses on the operational security and awareness of the internal situations of them. The cost efficiency has been proven by the system’s error-free performance offering a high availability during the 3-year operation.

Round 2
Reviewer 1 Report
We can see the effort you made in improving the article. However, there are some points which we still must point out for your deeper enhancement.
1. Revise the abstract, since the main contribution of the research is not clearly mentioned in the abstract.
2. The conclusion is lengthy. It should be further refined. Especially, we think that table 2 should be moved into the other portion of the article, not in the conclusion.
3. Please also add the other LMSs in the state of the art into the
into the comparison table, table 2.
Besides, we consider that you had better remove several unnecessary points in the article by focusing on the main topic of your study to attract the readers more. In general, the length of an article is normally under 25 pages. For example, there may be too many figures existed in your composition.

Author Response
We can see the effort you made in improving the article. However, there are some points which we still must point out for your deeper enhancement.
1. Revise the abstract, since the main contribution of the research is not clearly mentioned in the abstract.
-
First of all, we really appreciate your re-review. Following your comment, the contribution of the study has been added to Abstract while making modifications with the help of a native English speaker. Thus, we’d be most grateful if you will review our research work again. The additions and changes made are being highlighted in red.
Add) The major contribution of this study is that the design of the proposed LMS has been improved to provide a more efficient performance than the existing LMSs by surmounting the traffic overload problem often found in video services. This is achieved by utilizing a lesser number of servers and maintaining the balance of the loads. Also, the interface used for the system can be adaptable to most of the web servers as they support Java, Android, and HTML-based system. As a cloud-based LMS, this system has been tested for its efficiency and effectiveness for a period of three years during which the results have been satisfactory.
2. The conclusion is lengthy. It should be further refined. Especially, we think that table 2 should be moved into the other portion of the article, not in the conclusion.
-
Thank you for your appropriate comment. As you’ve pointed out, the conclusion section was unnecessarily lengthy so that we’ve refined it and moved a part of it to the other section highlighting it in blue. Additionally, we attempted to make Conclusion more meaningful.
3. Please also add the other LMSs in the state of the art into the into the comparison table, table 2.
-
We’ve further studied the latest LMS’s following your appropriate suggestion and they are being highlighted in red as below.
Besides, we consider that you had better remove several unnecessary points in the article by focusing on the main topic of your study to attract the readers more. In general, the length of an article is normally under 25 pages. For example, there may be too many figures existed in your composition.
-
We really appreciate your comments. As We’ve majored in computer science, there have been some mistakes. The theses pertaining to computer science can sometimes be longer as some descriptions and explanations have to be repeated [1-4]. We do agree with your comments saying that the contents are tedious and several unnecessary parts have been included. We’ve excluded some insignificant parts to reduce the length and tried to make the contribution of the research clearer while receiving the help of an English native speaker: Several contents seemingly irrelevant were deleted from ‘Algorithm’ and the code.
It is true that the implicative and meaningful words or descriptions can be an excellent example to every scholar. We’ve therefore read through some of the reputable studies [5-7] again to improve my paper. Again, thank you for your comments and a chance to revise the contents. The additions and revisions are being highlighted in red and your re-review will be highly appreciated.
[1] Hoon-Gi Lee, Jun-Ho Huh, A Cost-Effective Redundant Digital Excitation Control System and Test Bed Experiment for Safe Power Supply for Process Industry 4.0, Processes, MDPI, pp.1-29, (2018)
[2] Sangil Park, Jun-Ho Huh, Effect of Cooperation on Manufacturing IT Project Development and Test Bed for Successful Industry 4.0 Project: Safety Management for Security, Processes, MDPI, pp.1-31, (2018)
[3] Jun-Ho Huh, Big Data Analysis for Personalized Health Activities: Machine Learning Processing for Automatic Keyword Extraction Approach, Symmetry, MDPI, pp.1-30, (2018)
[4] Sangdo Lee, Jun-Ho Huh, An effective security measures for nuclear power plant using big data analysis approach, The Journal of Supercomputing, Springer US, pp.1-28, (2018)
[5] Vosoughi, S.; Roy, D.; Aral, S. The spread of true and false news online. Science, 2018, 359, 1146-1151.
[6] Jordan, M.I.; Mitchell, T.M. Machine learning: Trends, perspectives, and prospects. Science, 2015, 349, 255-260.
[7] LeCun, Y.; Bengio, Y.; Hinton, G. Deep learning. Nature, 2015, 521, 436-444.

Reviewer 2 Report
The information added in this revision is relevant and goes in line with the improvements i asked for. Nevertheless, I still have some remarks:
-Regarding the added statements on pages 29 and 30 - please include quantitative data to support the claims.
-A considerable percentage of this paper's contents is based on software-engineering details that add no novelty, nor value. Please focus the paper on the novel/relevant aspects of the approach. Just as an example: what Figure 7 provides in terms of novelty ? It depicts the use cases, which are rather neutral from a SE standpoint. This adds no value to the main rationale of the paper, which is supposedly focused in the "Efficient LMS Platform and Its Test Bed". Please focus the paper on the innovative aspects which provide added value.
Another suggestion, for future reply letters: use "we" instead of "I" - from what I understood the paper has more than one author.
Author Response
-
First of all, thank you for reading through our paper for review. Some of the contents have been revised with the assistance of a native English to correct the problematic parts. The corrections are being highlighted in red so that I’d appreciate very much if you will review the paper again.
Comments and Suggestions for Authors
The information added in this revision is relevant and goes in line with the improvements i asked for. Nevertheless, I still have some remarks:
-Regarding the added statements on pages 29 and 30 - please include quantitative data to support the claims.
-
While the part related to the comparisons between prices or performances are being explained on a CAPEX vs. OPEX basis, the parts we can disclose in the details have been additionally highlighted in red, excluding trade secrets. Thank you.
-A considerable percentage of this paper's contents is based on software-engineering details that add no novelty, nor value. Please focus the paper on the novel/relevant aspects of the approach. Just as an example: what Figure 7 provides in terms of novelty ? It depicts the use cases, which are rather neutral from a SE standpoint. This adds no value to the main rationale of the paper, which is supposedly focused in the "Efficient LMS Platform and Its Test Bed". Please focus the paper on the innovative aspects which provide added value.
-
We appreciate your comment. It is true that there were many unnecessarily lengthy parts in the manuscript. We’ve excluded some of the significant parts associated with software or network engineering while attempting to refine the contents by adding meaningful explanations to maintain consistency and completeness. Thus, we’d like to respectfully request your re-review. The changes and additions made are being highlighted in red whereas the part that has been moved is being indicated in blue.
Another suggestion, for future reply letters: use "we" instead of "I" - from what I understood the paper has more than one author.
-
Thank you for your comment. We’ve replaced “I” with “we” to avoid reader’s confusion.
